# Tex4D: Zero-shot 4D Character Texturing with Video Diffusion Models

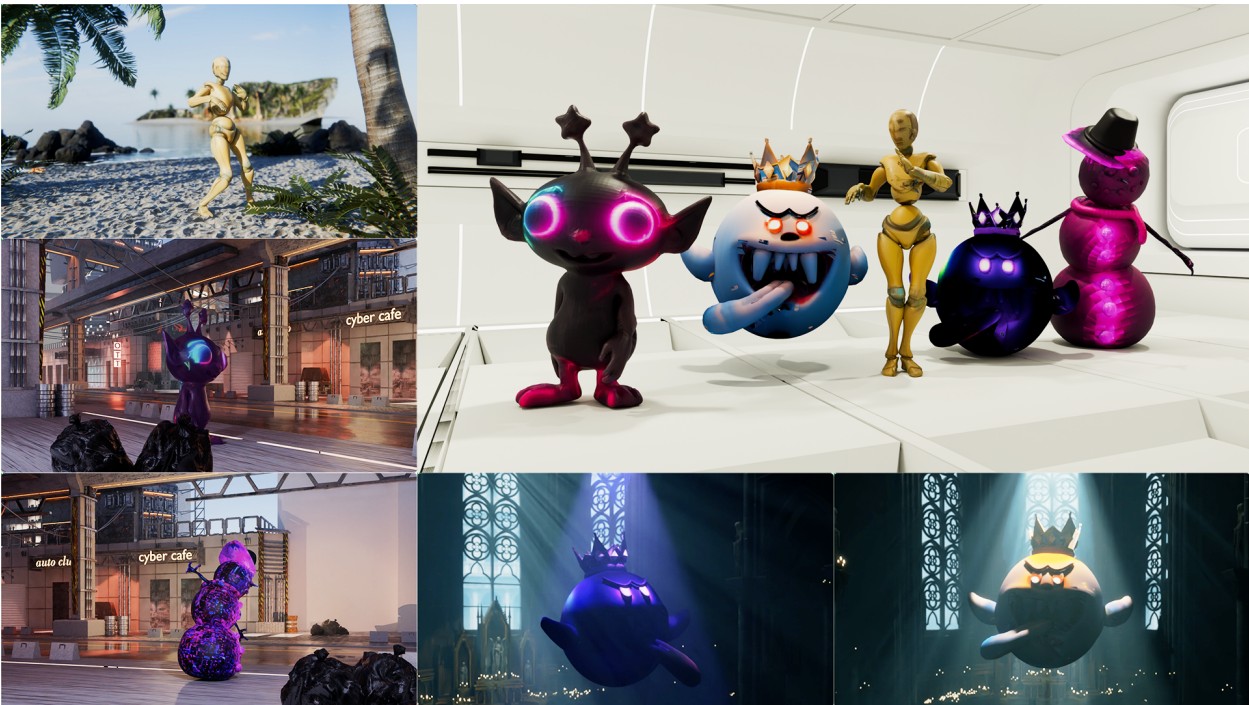

Figure 1. **Tex4D Application.** Our synthesized dynamic textures can be easily integrated into graphics pipelines.

## Abstract

3D meshes are widely used in movies, games, AR, and VR for their efficiency in animation and minimal memory footprint, leading to the creation of a large number of mesh sequences. However, creating dynamic textures for these mesh sequences to model the appearance transformations remains labor-intensive for professional artists. In this work, we present **Tex4D**, a zero-shot approach that creates multi-view and temporally consistent dynamic mesh textures by integrating the inherent 3D geometry knowledge with the expressiveness of video diffusion models. Given an untextured mesh sequence and a text prompt as inputs, our method enhances multi-view consistency by synchronizing the diffusion process across different views through latent aggregation in the UV space. To ensure temporal consistency, such as lighting changes, wrinkles, and appearance transformations, we leverage prior knowledge from a conditional video generation model for texture synthesis. Using the video diffusion model and the UV texture aggregation in a straightforward way leads to blurred results. We analyze the underlying causes and propose a simple yet effective modification to the DDIM sampling process to address this issue. Additionally, we introduce a reference latent texture to strengthen the correlation between frames during the denoising process. To the best of our knowledge, Tex4D is the first method specifically designed for 4D character texturing. Extensive experiments demonstrate its superiority in producing multi-view and multi-frame consistent dynamic textures for mesh sequences.

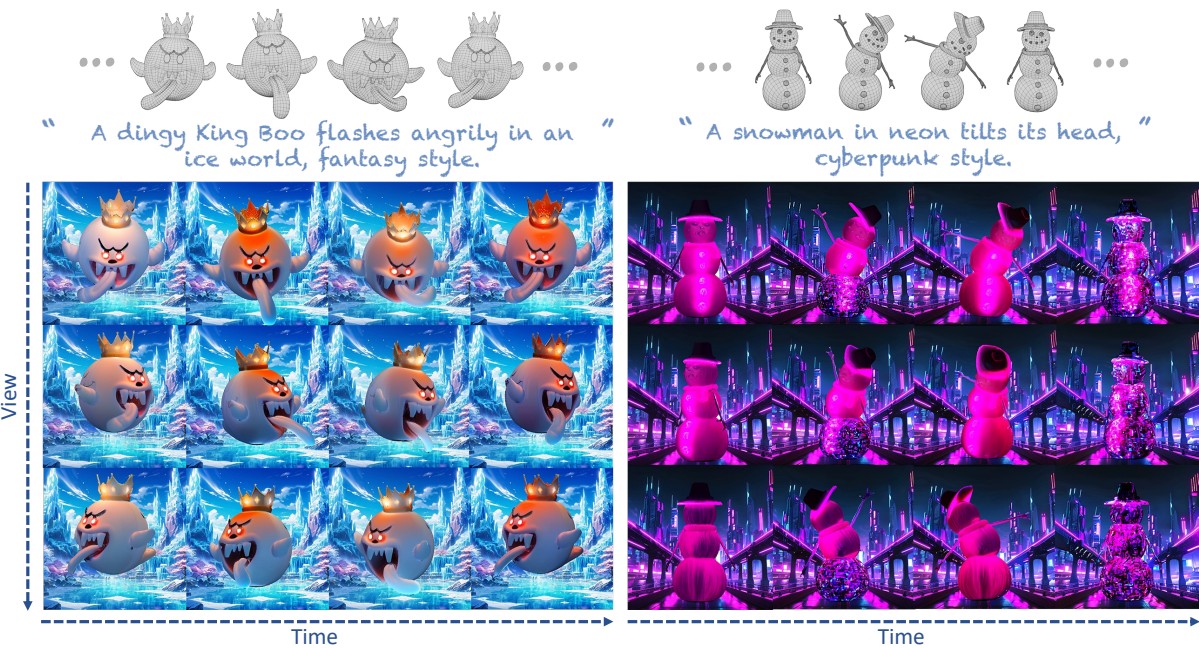

Figure 2. Given an untextured mesh sequence and a text prompt as inputs (top), **Tex4D** generates multi-view, dynamic textures. We show renderings of the textured meshes from three views and four timestamps (bottom).

# 1 Introduction

3D meshes are widely used in computer-aided design (CAD), animation, and computer graphics due to their low memory footprint and efficiency in animation. Visual artists, game designers, and movie creators build numerous animated mesh sequences for visual applications. However, creating vivid videos involves complex post-processing steps, such as creating dynamic textures for appearance transformations, as shown in Fig. 1. These steps are labor-intensive and require expertise from artists.

On the other hand, recent advancements in generative models have democratized content creation and demonstrated impressive performance in image and video synthesis. For instance, video generation models (Ho et al., 2022; Esser et al., 2023; Li et al., 2023; He et al., 2022; Yu et al., 2023a; Zhou et al., 2022; Hong et al., 2022; Yang et al., 2024; Zhang et al., 2023b; Xing et al., 2023; Chen et al., 2023c; 2024) trained on large-scale video datasets (Bain et al., 2021; Schuhmann et al., 2021) allow users to create realistic video clips from various modalities such as text prompts, images, or geometric conditions. However, these text-to-video generation models, which are trained solely on 2D data, often struggle with spatial consistency when applied to multi-view image generation (Tang et al., 2023; Shi et al., 2023b; Liu et al., 2023a; Weng et al., 2023; Long et al., 2023; Shi et al., 2023a; Kwak et al., 2023; Tang et al., 2024; Voleti et al., 2024) or 3D object texturing (Cao et al., 2023; Liu et al., 2023b; Richardson et al., 2023; Huo et al., 2024).

To address these limitations, two main approaches have been developed. One approach (Richardson et al., 2023; Chen et al., 2023b; Cao et al., 2023) focuses on resolving multi-view inconsistency in static 3D object texturing by synchronizing multi-view image diffusion processes. While these methods produce multi-view consistent textures for static 3D objects, they do not address the challenge of generating dynamically changing textures for mesh sequences (e.g., the lighting effects shown in Fig. 1). Another approach (Guo et al., 2023a; Lin et al., 2024; Peng et al., 2024) aims to generate video clips based on the rendering (e.g., depth, normal or UV maps) of an untextured mesh sequence. To encourage temporal consistency, these methods modify the attention mechanism in 2D diffusion models and utilize inherent correspondences in a mesh sequence to facilitate feature synchronization between frames. Although these techniques can be adapted for multi-view image generation by treating camera pose movement as temporal motion, they usually produce inconsistent 3D texturing due to insufficient exploitation of 3D geometry priors.

In this paper, we introduce a novel task: 4D character texturing. Given an animated untextured 3D mesh sequence and a text prompt, our goal is to generate dynamic textures that are both temporally and multi-view consistent. We aim to texture 3D mesh sequences by capturing temporal variations through dynamic textures – a fundamental task in film-making and game design. Unlike existing works, we fully leverage 3D geometry knowledge from the mesh sequence to enforce multi-view consistency. Specifically, we develop a method that synchronizes the diffusion process from different views through latent aggregation in the UV space. To ensure temporal consistency, we employ prior knowledge from a conditional video generation model for texture sequence synthesis and introduce a reference latent texture to enhance frame-to-frame correlations during the denoising process. However, naively integrating the UV texture aggregation into the video diffusion process causes the variance shift problem, leading to blurry results. To resolve this issue, we propose an effective modification to the DDIM (Song et al., 2020) sampling process. Our method is a plug-and-play module designed for video diffusion models. The textured meshes can be rendered from any camera view, thus supporting various applications in content creation. Our key contributions are:

- We present **Tex4D**, a zero-shot pipeline for generating high-fidelity dynamic textures that are temporally and multi-view consistent, utilizing video diffusion models and mesh sequence controls.
- To leverage priors from existing video diffusion models, we develop an effective modification to the DDIM sampling process to address the variance shift issue caused by multi-view texture aggregation and design a background learning module.
- We introduce a reference UV blending mechanism to establish correlations during the denoising steps, addressing self-occlusions, and synchronizing the diffusion process in invisible regions.
- Our method demonstrates comparable if not superior performance to various state-of-the-art baselines.

## 2   Related Work

**Video Stylization and Editing.** Video diffusion models have shown remarkable performance in the field of video generation. These models learn motions and dynamics from large-scale video datasets using 3D-UNet to create high-quality, realistic, and temporally coherent videos. Although these approaches show compelling results, the generated videos lack fine-grained control, inhibiting their application in stylization and editing. To solve this issue, inspired by ControlNet (Zhang et al., 2023a), SparseCtrl (Guo et al., 2023a) trains a sparse encoder from scratch using frame masks and sparse conditioning images as input to guide the video diffusion model. CTRL-Adapter (Lin et al., 2024) proposes a trainable intermediate adapter to connect the features between ControlNet and video diffusion models.

Meanwhile, (Tumanyan et al., 2023) observed that the spatial features of text-to-image (T2I) models play an influential role in determining the structure and appearance, Text2Video-Zero (Khachatryan et al., 2023) uses a frame-warping method to animate the foreground object by T2I models and (Wu et al., 2023; Ceylan et al., 2023; Qi et al., 2023) propose utilizing self-attention injection and cross-frame attention to generate stylized and temporally consistent video using DDIM inversion (Song et al., 2020). Subsequently, numerous works (Zhang et al., 2023c; Cai et al., 2024; Yang et al., 2023; Geyer et al., 2023; Eldesokey & Wonka, 2024) generate temporally consistent videos utilizing T2I diffusion models by spatial latent alignment without training. However, the synthesized videos usually show flickerings due to the empirical correspondences, such as feature embedding distances and UV maps, which are insufficient to express the continuous content in the latent space. Another line of work (Singer et al., 2022; Bar-Tal et al., 2022; Blattmann et al., 2023; Xu et al., 2024; Guo et al., 2023b) is to train additional modules on large-scale video datasets to construct feature mappings, for example, Text2LIVE (Bar-Tal et al., 2022) applies test-time training with the CLIP loss, and MagicAnimate (Xu et al., 2024) introduced an appearance encoder to retain intricate clothes details.

**Texture Synthesis.** With the rapid development of foundation models, researchers have focused on applying their generation capability and adaptability to simplify the process of designing textures and reduce the expertise required. To incorporate the result 3D content with prior knowledge, earlier works (Khalid et al., 2022; Michel et al., 2021; Chen et al., 2022) jointly optimize the meshes and textures from scratch with the simple semantic loss from the pre-trained CLIP (Radford et al., 2021) to encourage the 3D alignment between the generated results and the semantic priors. However, the results show apparent artifacts and distortion because the semantic feature cannot provide fine-grained supervision during the generation of 3D content.

DreamFusion (Poole et al., 2022) and similar models (Lin et al., 2023; Wang et al., 2023; Po & Wetzstein, 2024; Metzer et al., 2022; Chen et al., 2023a) distill the learned 2D diffusion priors from the pre-trained diffusion models (Rombach et al., 2021) to synthesize the 3D content by Score Distillation Sampling (SDS). These methods render 2D projections of the 3D asset parameters and compare them against reference images, iteratively refining the 3D asset parameters to minimize the discrepancy of the target distribution of 3D shapes learned by the diffusion model. Although these approaches enable people without expertise to generate detailed 3D content by textual prompt, their results are typically over-saturated and over-smoothed, hindering their application in actual cases. Another line of optimization-based methods (Yu et al., 2023b; Zeng et al., 2024; Bensadoun et al., 2024) turned to fuse 3D shape information, such as vertex positions, depth maps, and normal maps, with the pre-trained diffusion model by training separate modules on 3D datasets. Still, they require a specific UV layout process to achieve plausible results.

Recently, TexFusion (Cao et al., 2023) and numerous zero-shot methods (Liu et al., 2023b; Richardson et al., 2023; Huo et al., 2024) have shown significant success in generating globally consistent textures without additional 3D datasets. Based on depth-aware diffusion models, they sequentially inpaint the latents in the UV domain to ensure the spatial consistency of latents observed across different views. Then, they decode the latents from multiple views and finally synthesize the RGB texture through back projection.

However, these methods generate static 3D assets and overlook temporal changes in visual presentations, such as videos. To our knowledge, this is the first approach to synthesize multi-view dynamic textures for mesh sequences, enabling appearance transformations.

## 3 Preliminaries

**Video Diffusion Prior.** In this paper, we adopt CTRL-Adapter (Lin et al., 2024) as our prior model to provide dynamic information. CTRL-Adapter aims to adapt a pre-trained text-to-video diffusion model to conditions for various types of images, such as depth or normal map sequences. The key idea behind CTRL-Adapter is to leverage a pre-trained ControlNet (Zhang et al., 2023a) and to align its latents with those of the video diffusion model through a learnable mapping module. Intuitively, the video diffusion model generates temporally consistent video frames that capture dynamic elements like character motions and lighting, while the ControlNet further enhances this capability by allowing the model to condition on geometric information, such as depth and normal map sequences. This makes CTRL-Adapter particularly effective in providing a temporally consistent texture prior to our 4D character texturing task. Specifically, we leverage the depth-conditioned CTRL-Adapter model. Given a sequence of depth images denoted as $\{D_1, ..., D_K\}$ and a text prompt $\mathcal{P}$, CTRL-Adapter (denoted as $\mathcal{C}$) synthesizes a RGB frame sequence $F$ by $F = \mathcal{C}(\{D_1, ..., D_K\}, \mathcal{P})$.

**DDIM Sampling.** DDIM (Song et al., 2020) is a widely used sampling method in diffusion models due to its superior efficiency and deterministic nature compared to DDPM (Ho et al., 2020). To enhance numerical stability and prevent temporal color shifts in video diffusion, numerous models (Zhang et al., 2023b; Ho et al., 2022) employ a learning-based sampling technique known as v-prediction (Salimans & Ho, 2022). At each denoising step, the sampling process for the latents (denoted as $z_t$) can be described as follows:

$$z_{t-1} = \sqrt{\alpha_{t-1}} \cdot \hat{z}_0(z_t) + \sqrt{1 - \alpha_{t-1}} \cdot \epsilon_\theta(z_t),$$
$$\hat{z}_0(z_t) = \frac{z_t - \sqrt{1 - \alpha_t} \cdot \epsilon_{\mathrm{UNet}}}{\sqrt{\alpha_t}}, \quad \epsilon_\theta(z_t) = \epsilon_{\mathrm{UNet}}, \tag{1}$$

where $\alpha_t$ is the noise variance at time step $t$, $\epsilon_{\mathrm{UNet}}$ is the estimated noise from the U-Net denoising module, which is expected to follow $\mathcal{N}(0, \mathcal{I})$, and $\hat{z}_0(z_t)$ denotes the predicted original sample (i.e., the latents at timestep 0). After the v-parameterization, the predicted original sample $\hat{z}_0(z_t)$ and the predicted epsilon $\epsilon_\theta(z_t)$ are computed as follows:

$$\hat{z}_0(z_t) = \sqrt{\alpha_t} \cdot z_t - \sqrt{1 - \alpha_t} \cdot \epsilon_{\mathrm{UNet}}, \quad \epsilon_\theta(z_t) = \sqrt{\alpha_t} \cdot \epsilon_{\mathrm{UNet}} + \sqrt{1 - \alpha_t} \cdot z_t. \tag{2}$$

We propose an enhanced DDIM sampling process (Sec. 4.3) in video diffusion models, along with a multi-view consistent texture aggregation mechanism (Sec. 4.2) to synthesize consistent 4D textures.

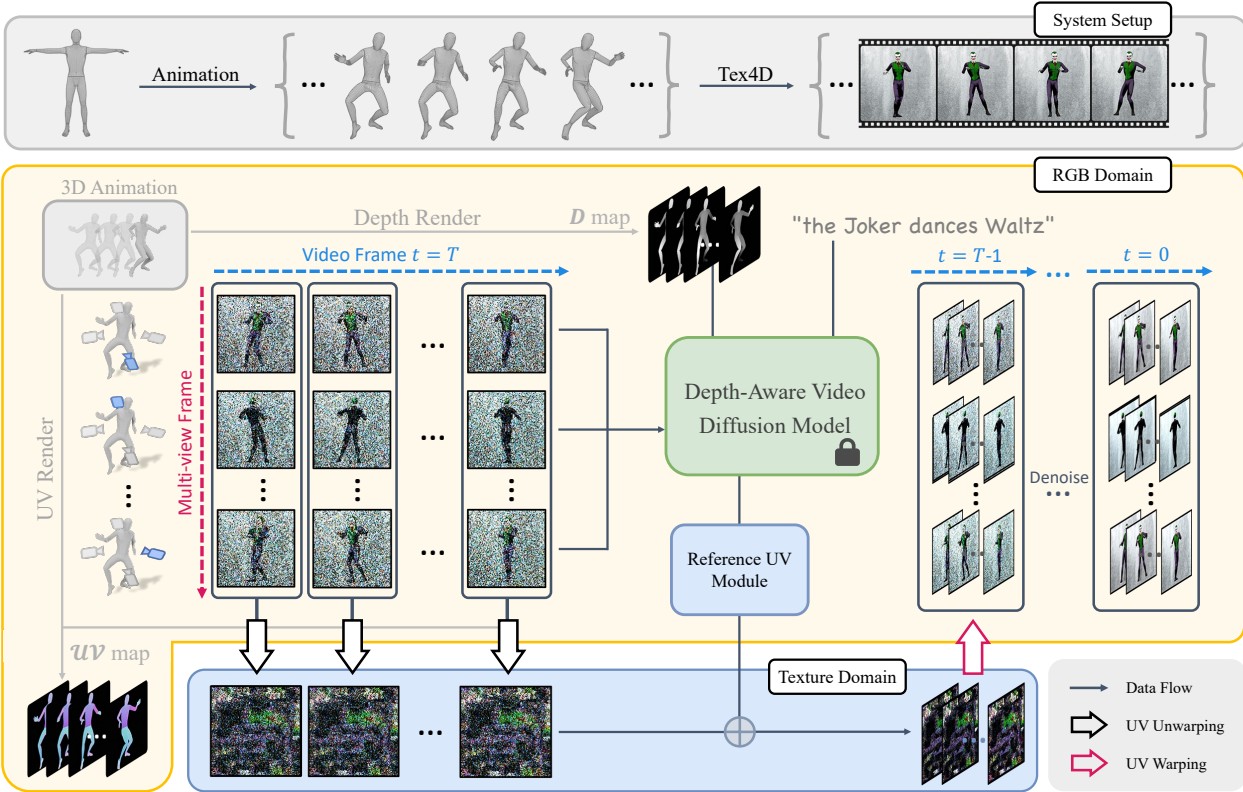

Figure 3. **Overview.** Given a mesh sequence and a text prompt as inputs, Tex4D generates a UV-parameterized texture sequence that is both globally and temporally consistent. At each diffusion step, latent views are aggregated into UV space, followed by multi-view latent texture diffusion to ensure global consistency. To maintain temporal coherence and address self-occlusions, a Reference UV Blending module is applied at each step. Finally, the latent textures are back-projected and decoded to produce RGB textures for each frame.

## 4  Method

Given an untextured mesh animation and a text prompt, our goal is to generate a multi-view and multi-frame consistent texture sequence for each mesh that aligns with both the text description and motion cues while capturing the dynamics from video diffusion models.

To optimize computational efficiency, we uniformly sample $K$ key frames from the video and synthesize textures for these keyframes. Textures for the remaining frames are then generated by interpolating the key frame textures. Formally, given $K$ animated meshes at the keyframes ($\{M_1, ..., M_K\}$), along with a text description $\mathcal{P}$, our method produces temporally and spatially consistent UV maps denoted as $\{UV_1, ..., UV_K\}$, in a zero-shot manner.

Previous texture generation methods  (Richardson et al., 2023; Chen et al., 2023b; Cao et al., 2023) typically inpaint and update textures sequentially using pre-defined camera views in an incremental manner. However, these approaches rely on view-dependent depth conditions and lack global spatial consistency, often resulting in visible discontinuities in the assembled texture map. This issue arises from error accumulation during the autoregressive view update process, as noted by Bensadoun et al. (2024). To resolve these issues, rather than processing each view independently, recent methods (Liu et al., 2023b) propose to generate multi-view textures simultaneously through diffusion. In this work, we similarly leverage the UV space as an intermediate representation to ensure multi-view consistency.

### 4.1  Overview

As shown in Fig. 3, given a sequence of $K$ meshes, we start by rendering the mesh at $V$ predefined, uniformly sampled camera poses to obtain multi-view depth images (denoted as $\{D_{1,1}, ..., D_{1,K}, D_{2,1} ..., D_{V,K}\}$), which

serve as the geometric conditions. To generate textures for each mesh, we initialize $V \times K$ noise images sampled from a Normal distribution (denoted as $\{z^{1,1}, ..., z^{1,K}, z^{2,1}, ..., z^{V,K}\}$). Additionally, we initialize an extra noise map sequence $\{z_b^1, ..., z_b^K\}$ for the backgrounds learning. This noise map corresponds to the texture of a plane mesh that is composited with the foreground object at each diffusion step (See Sec. 4.3). Next, for each view $v \in \{1, ..., V\}$, we apply the video diffusion model (Lin et al., 2024) discussed in Sec. 3 to simultaneously denoise all latents and obtain multi-frame consistent images as $\{I^{1,v}, ..., I^{K,v}\} = \mathcal{C}(\{D_{1,v}, ..., D_{K,v}\}, \mathcal{P})$, where $\mathcal{P}$ is the provided text prompt. Finally, we un-project and aggregate all denoised multi-view images for each mesh to formulate temporally consistent UV textures.

Applying the video diffusion model independently to each camera view often results in multi-view inconsistencies. Inspired by (Liu et al., 2023b; Huo et al., 2024; Zhang et al., 2024), we aggregate the multi-view latents of each mesh in the UV space to merge observations across different views at each denoising step, and then render latent from the latent texture to ensure multi-view consistency. Furthermore, we composite the rendered foreground latents with the background latents at each diffusion step (discussed in Sec. 4.2), which is essential to exploit prior in the video diffusion model (see Fig. 15). Nonetheless, such a simple aggregation method introduces blurriness in the final results. In Sec. 4.3, we analyze the underlying causes and propose a simple yet effective method to enhance the denoising process. Additionally, we create a reference UV to handle self-occlusions and further improve temporal consistency in Sec. 4.4.

## 4.2 Multi-view Latents Aggregation in the UV Space

We describe the aggregation of multi-view latents in the UV space. For frame $k \in \{1, ..., K\}$, we aggregate the multi-view latents $\{z^{1,k}, ..., z^{V,k}\}$ in the UV space by:

$$\mathcal{T}\left(z^k\right) = \frac{\sum_{v=1}^{V} \mathcal{R}^{-1}(z^{v,k}, c_v) \odot \cos(\theta^v)^\alpha}{\sum_{v=1}^{V} \cos(\theta^v)^\alpha}, \tag{3}$$

where $\mathcal{R}^{-1}$ represents the inverse rendering operator that un-projects the latents to the UV space, thus $\mathcal{R}^{-1}(z^{v,k}, c_v)$ produces a partial latent UV texture from view $v$, $\cos(\theta^v)$ is the cosine map buffered by the geometry shader, recording the cosine value between the view direction and the surface normal for each pixel, $\alpha$ is a scaling factor, and $c_v$ denotes one of the predefined cameras. After multi-view latents aggregation, we obtain multi-view consistent latents by rendering the aggregated UV latent map using $\tilde{z}^{v,k} = \mathcal{R}\left(\mathcal{T}^k; c_v\right)$, where $\mathcal{R}$ is the rendering operation.

## 4.3 Multi-frame Texture Generation

The aggregation process discussed above yields multi-view consistent latents $\{\tilde{z}^{v,k}\}$ for the denoising steps. However, this simple aggregation and projection strategy leads to a blurry appearance, as shown in Fig. 8(b). This issue arises primarily because the aggregation process depicted in Eq. 3 derails the DDIM denoise process. Specifically, the estimated noise $\epsilon_\theta(z_t)$ for each step in Eq. 1 is expected to follow $\mathcal{N}(0, \mathcal{I})$, but Eq. 3 indicates that after aggregating multi-view latents, the expected norm of variance of the noise distribution would be less than $\mathcal{I}$. We denote this as the "variance shift" issue caused by the texture aggregation.

To resolve this issue, we rewrite the estimated noise $\epsilon_{\text{UNet}}$ as the combination of the t-step latent $z_t$ and the estimated latent $\hat{z}_0(z_t)$ at step 0. The v-paramaterized predicted epsilon $\epsilon_\theta(z_t)$ in Eq. 2 can be equivalently expressed as:

$$\begin{aligned}
\epsilon_{\text{UNet}} &= \left(\sqrt{\alpha_t} \cdot z_t - \hat{z}_0(z_t)\right) / \sqrt{1 - \alpha_t} \\
\epsilon_\theta(z_t) &= \sqrt{\alpha_t} \cdot \epsilon_{\text{UNet}} + \sqrt{1 - \alpha_t} \cdot z_t \\
&= \sqrt{\frac{\alpha_t}{1 - \alpha_t}} \cdot \left(\sqrt{\alpha_t} z_t - \hat{z}_0(z_t)\right) + \sqrt{1 - \alpha_t} \cdot z_t.
\end{aligned} \tag{4}$$

In practice, we carry out this denoising technique in the UV space. Specifically, we first compute the original texture map (i.e., texture map at step 0, denoted as $\hat{\mathcal{T}}_0$) by aggregating the predicted original multi-view image latents through Eq. 3. The noisy latent texture map at time step $t$ (denoted as $\mathcal{T}_t$) can be similarly

computed. We denoise in step $t$ by:

$$\mathcal{T}_{t-1} = \sqrt{\alpha_{t-1}} \cdot \hat{\mathcal{T}}_0 + \sqrt{1 - \alpha_{t-1}} \cdot \left( \sqrt{\frac{\alpha_t}{1 - \alpha_t}} \cdot (\sqrt{\alpha_t}\mathcal{T}_t - \hat{\mathcal{T}}_0) + \sqrt{1 - \alpha_t} \cdot \mathcal{T}_t \right). \tag{5}$$

Through experimentation, we observe that background optimization plays a crucial role in fully exploiting the prior within the video diffusion model. As shown in Fig. 8(c), using a simple white background produces blurry results. This may stem from a mismatch between the white-background images and the training dataset, which likely contains fewer such examples, affecting the denoising process. To resolve this issue, we compute the final latents as the combination of the foreground latent $\tilde{z}_{t-1}$ projected from the aggregated UV latents and the residual background latent $z_{b,t-1}$ denoised by diffusion models. Specifically, we composite the estimated latents in the $t-1$ step as follows:

$$z_{t-1} = \tilde{z}_{t-1} \odot \mathcal{M}_{\text{fg}} + z_{b,t-1} \odot (1 - \mathcal{M}_{\text{fg}}), \quad \tilde{z}_{t-1}, \mathcal{M}_{\text{fg}} = \mathcal{R}\left(\mathcal{T}_{t-1}; c_v\right), \tag{6}$$

where $\mathcal{M}_{\text{fg}}$ represents the foreground mask of the mesh, and $\mathcal{R}$ is the rendering operation.

To summarize, we start with $K \times (V + 1)$ randomly initialized noise maps sampled (i.e., $K \times V$ maps for foreground, $K$ maps for background) and denoise them into images simultaneously. At each denoising step $t$ with frame $k$, we derive the estimated noises $\{\epsilon_{t-1}^{1,k}, \ldots, \epsilon_{t-1}^{V,k}\}$ using the video diffusion model and calculate the estimated original latent $\{\hat{z}_0^{1,k}, \ldots, \hat{z}_0^{V,k}\}$ by Eq. 1. Then, we use Eq. 3 to aggregate the latents onto UV space. Next, we utilize Eq. 5 to take the diffusion step in the UV space, and render the synchronized latents $\{\tilde{z}_{t-1}^{1,k}, \ldots, \tilde{z}_{t-1}^{V,k}\}$ from latent UVs $\{\mathcal{T}_{t-1}^1, \ldots, \mathcal{T}_{t-1}^K\}$ to ensure multi-view consistency. Finally, we composite the denoised latent with the latents at step $t-1$ according to foreground masks by Eq. 6. Algorithm 1 in the supp shows the full procedure.

## 4.4 Reference UV Blending

Although the video diffusion model maintains temporal consistency for latents from each view, this consistency may sometimes decrease after aggregation in the texture domain. This issue primarily stems from the view-dependent nature of the depth conditions and the limited resolution of latents, which can lead to distortions when features from different camera angles are combined onto the UV texture. Additionally, self-occlusion during animation results in information loss in invisible regions.

To address these challenges, we propose a reference UV map to enhance correlations between latent textures across frames. Specifically, the reference UV map is constructed by sequentially combining latent textures over time, with each new texture filling only the empty texels of the reference UV map. Each texture is blended using the reference UV $\mathcal{T}_{\mathcal{UV}}$ with a mask $\mathcal{M}_{\mathcal{UV}}$ that labels the visible region:

$$\mathcal{T}_t^k = \left((1 - \lambda) \cdot \mathcal{T}_t^k + \lambda \cdot \mathcal{T}_{\mathcal{UV}}\right) \odot \mathcal{M}_{\mathcal{UV}}^k + \mathcal{T}_{\mathcal{UV}} \odot \left(1 - \mathcal{M}_{\mathcal{UV}}^k\right) \tag{7}$$

where $\lambda$ is the blending weight for the reference UV in the visible region, while the invisible region is simply replaced with the reference texture. We empirically set the blending weight to 0.2.

## 5 Experiments

**Datasets.** We source our datasets from two primary repositories: human motion diffusion outputs, the Mixamo and Sketchfab websites. We employ the text-to-motion diffusion model (Tevet et al., 2023) to compare our approach with LatentMan (Eldesokey & Wonka, 2024). For comparison with Generative Rendering (Cai et al., 2024), we obtain animated characters from Mixamo. Specifically, we first use Blender Community (2024) to extract meshes, joints, skinning weights, and animation data from the FBX files. Then, we apply linear blend skinning to animate the meshes. We utilize XATLAS to parameterize the mesh UVs.

**Baselines.** To the best of our knowledge, no existing studies tackle dynamic texture generation. We thus compare with 8 SOTA methods including Text-to-4D methods, video stylization methods and video generation methods with various control mechanisms. SV4D (Xie et al., 2024) and L4GM (Ren et al., 2024)

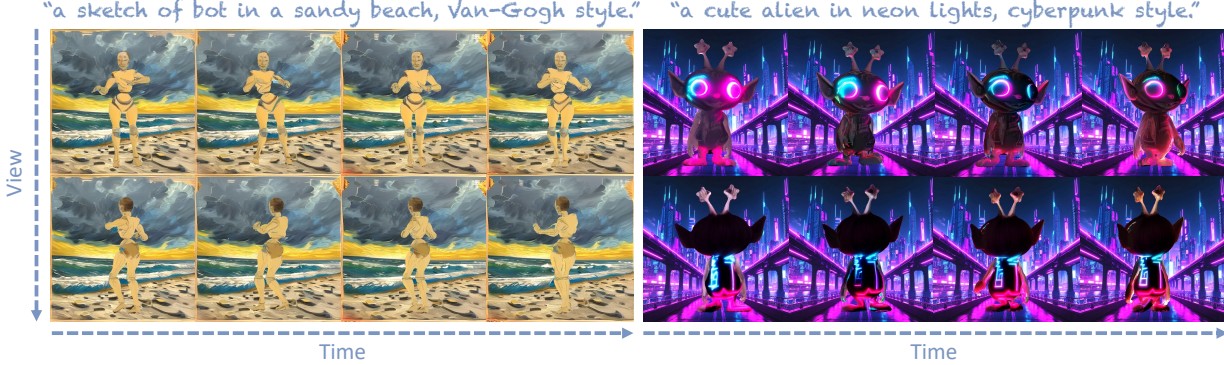

Figure 4. **Qualitative Results.** Our method generates multi-view consistent dynamic textures with a diverse set of styles and prompts. Zoom in to view the details. More results are provided in the supplementary material.

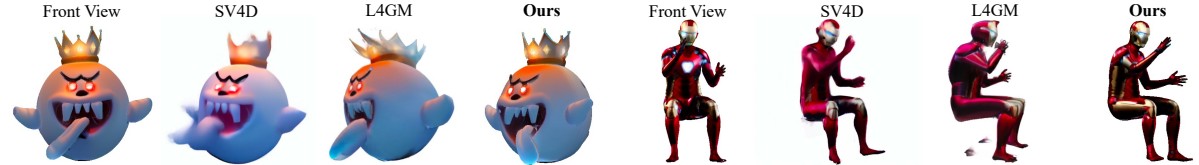

Figure 5. **Qualitative Comparison with Text-to-4D Methods.** Our method generates multi-view consistent results compared with Text-to-4D methods as our approach fully utilizes the geometry information of the meshes.

are text-to-4D methods, both taking a front view as the input. PnP-Diffusion (Tumanyan et al., 2023) is an image stylization method that guides the generation with DDIM features. We extend the method on a frame-by-frame basis for comparison, aligning with previous work (Geyer et al., 2023). Built upon cross-frame attention, Text2Video-Zero (Khachatryan et al., 2023) guides the video by warping latents to enhance video dynamics implicitly. We leverage its official extension for comparison.TokenFlow (Geyer et al., 2023), Generative Rendering (Cai et al., 2024), and LatentMan (Eldesokey & Wonka, 2024) establish latent correspondences through nearest neighbor and DensePose features. Gen-1 (Esser et al., 2023) is a video-to-video model that transforms the untextured mesh renders into stylized outputs. Given the lack of source code for Generative Rendering, we utilize the experimental results presented in their video demos for qualitative comparison. Additionally, we compare our method with the texture generation method Text2Tex (Chen et al., 2023b).

**Evaluation Metrics.** Quantitatively evaluating multi-view consistency and temporal coherence remains challenging. We conduct a user study to assess overall performance, including appearance quality, spatio-temporal consistency, and prompt fidelity based on human preferences. Additionally, we measure video-level multi-view temporal coherence using Fréchet Video Distance (FVD) (Unterthiner et al., 2018) following (Li et al., 2024; Xie et al., 2024), along with a CLIP-based Consistency Score following (Liu et al., 2023b).

## 5.1 Qualitative Results

We present qualitative evaluation in Fig. 5 and Fig. 6. Text-to-4D methods fail to render plausible results as they entangle geometry and appearance. Generative Rendering, TokenFlow, and Text2Video-Zero rely on T2I diffusion models with cross-frame attention mechanisms and exhibit noticeable flickering. This issue stems from misalignment between synthesized frames and latents, where temporally consistent latents may be decoded into inconsistent RGB frames. In contrast, Tex4D interpolates frames between keyframe textures directly in RGB space, bypassing inconsistencies caused by latent manipulation. PnP-Diffusion edits frames independently and generates detailed and sophisticated appearances but suffers from poor spatio-temporal consistency due to the loss of temporal correlations in the latent space. While Gen-1 produces high-quality videos, it fails to maintain multi-view consistency.

Furthermore, we present multi-view results showcasing a variety of styles and prompts in Fig. 4. Our method, driven by video diffusion models, effectively accounts for the styles and captures temporal variations over time. As shown in Fig. 7, Tex4D effectively handles the invisible regions compared with the traditional

Table 1. **Quantitative evaluation**. We present metric values and a comparison of the percentage and statistics of user preference for our approach against other methods. Our method shows the best spatio-temporal consistency as measured by the FVD (Unterthiner et al., 2018) and Consistency Score (Liu et al., 2023b). Users consistently favored Tex4D over all baselines.

| Method | FVD (↓) | Cons. Score (↑) | Appearance Quality | Spatio-temporal Consistency | Consistency with Prompt |
|---|---|---|---|---|---|
| Text2Video-Zero | 3078.94 | 86.80 | 89.33% (2.13) | 91.78% (2.17) | 91.55% (2.63) |
| PnP-Diffusion | 1390.04 | 86.48 | 86.42% (2.92) | 87.18% (2.79) | 89.74% (2.88) |
| TokenFlow | 1330.43 | 87.35 | 92.31% (3.08) | 86.84% (4.04) | 93.42% (3.08) |
| Gen-1 | 3114.26 | 81.74 | 70.27% (4.46) | 75.00% (3.33) | 77.78% (4.63) |
| LatentMan | 2811.23 | 86.50 | 86.57% (3.63) | 86.57% (3.88) | 81.82% (3.75) |
| Ours | **1303.14** | **95.35** | -     (**4.69**) | -     (**4.84**) | -     (**4.82**) |

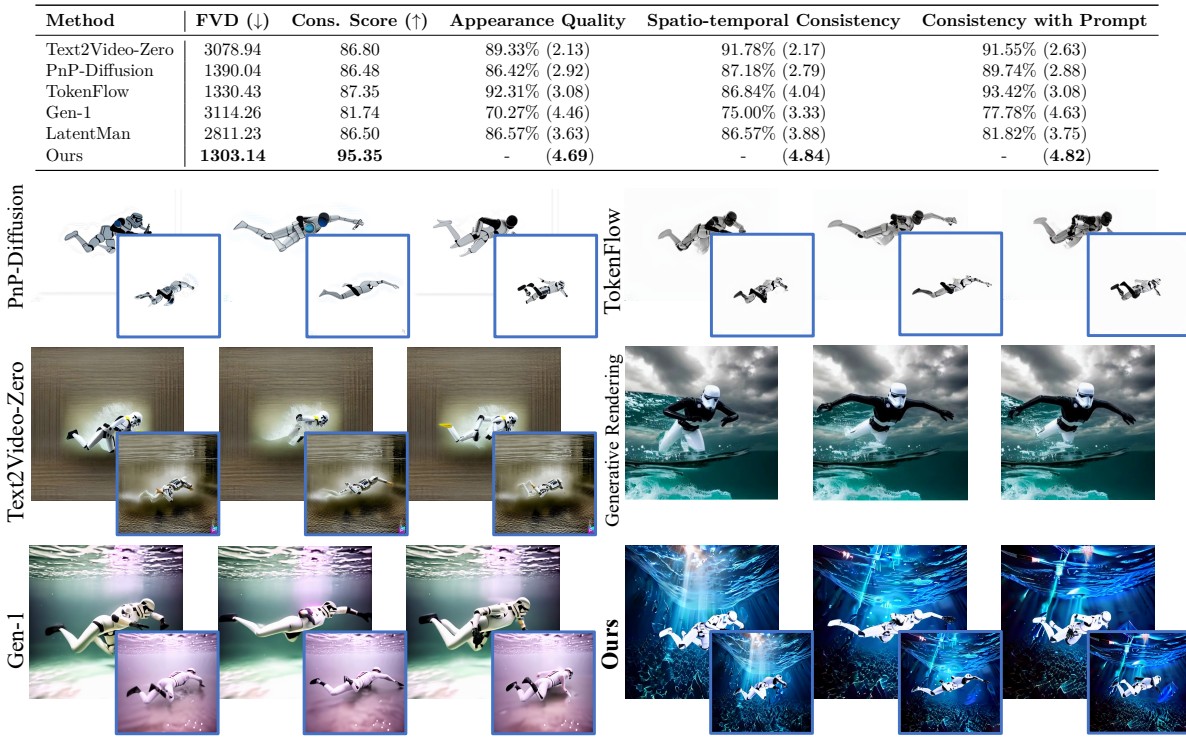

"a Stormtrooper is Swimming"

Figure 6. **Qualitative comparisons of multi-view video generation.** We compare our method against PnP-diffusion (Tumanyan et al., 2023), TokenFlow (Geyer et al., 2023), Text2Video-Zero (Khachatryan et al., 2023), Generative Rendering (Cai et al., 2024) (from their video demo), and Gen-1 (Esser et al., 2023). We generate videos in the front view and the side view (blue box) on Mixamo dataset. Our method generates vivid videos that align with the textual prompts while preserving spatial consistency.

texture generation method Text2Tex (Chen et al., 2023b), which also fails to model dynamics. We provide analysis and comparison of Tex4D with textured mesh animation in Sec D.2 in the supp.

## 5.2 Quantitative Evaluation

To quantitatively assess the effectiveness of our proposed method, we follow prior research (Eldesokey & Wonka, 2024; Geyer et al., 2023; Esser et al., 2023) and conduct a comprehensive A/B user study. Our study involved 67 participants who provided a total of 1104 valid responses based on six different scenes drawn from six previous works, with each scene producing videos from two different views. During each evaluation, participants were presented with rendered meshes and depth conditions viewed from two angles, serving as motion references. They were shown a pair of videos: one generated by our approach and the other from a baseline method. Partic-

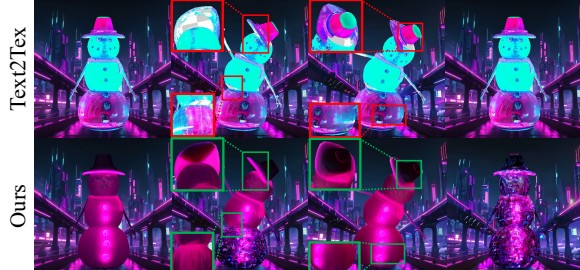

Figure 7. **Comparison with Text2Tex.** Texture generation method (Chen et al., 2023b) shows empty texels in invisible regions and fails to model dynamics.

ipants were asked to select the method that exhibited superior performance in three criteria: 1) appearance quality, 2) spatial and temporal consistency, and 3) fidelity to the prompts. Table 1 summarizes the preference percentage and statistics (ranging from 1 to 5) of our method over other methods. Our method significantly surpasses other methods by a large margin. In addition, our method achieves the best FVD (Unterthiner

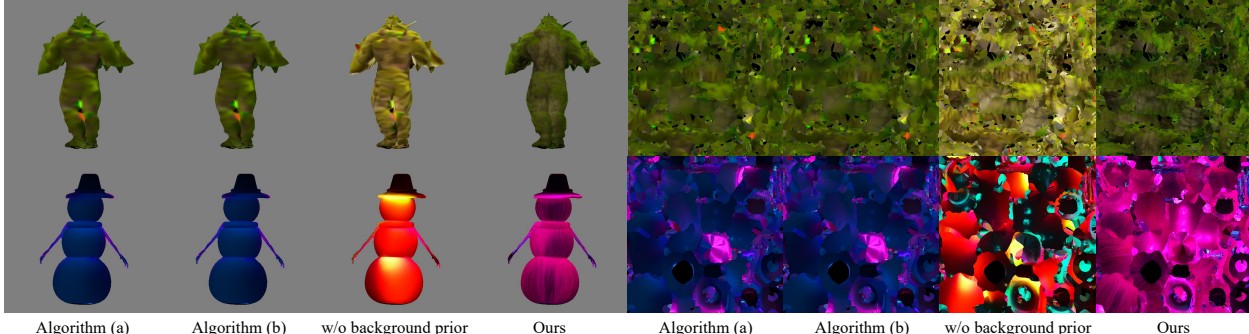

Algorithm (a)    Algorithm (b)    w/o background prior    Ours    Algorithm (a)    Algorithm (b)    w/o background prior    Ours

Figure 8. **Comparisons of multi-view denoising algorithms and ablation on background priors.** (a) The simple multi-view diffusion algorithms by Eq. 1, and (b) aggregation of $z_{t-1}$ result in a blurry appearance compared to (d) our results. Without background priors, our approach fails to generate plausible video clips.

et al., 2018) and Consistency Score (Liu et al., 2023b), which demonstrates better multi-view consistency in generated video clips.

### 5.3 Ablation Study

**Ablation for texture aggregation.** In Fig. 8 (a) and (b), we present two alternative texture aggregation methods. In Fig. 8 (a), we un-project $\hat{z}_0(z_t)$ and $\epsilon_\theta(z_t)$ into UV space for aggregation. In Fig. 8 (b), we map $z_{t-1}$ to the UV space. Both of these approaches show inferior results compared to our method, which verifies the effectiveness of the proposed texture aggregation algorithm.

**Ablation for UV blending module.** In Sec. 4.4, we propose a reference UV blending schema to resolve the temporal inconsistency caused by latent aggregation. To validate the effectiveness of this mechanism (See Sec. 4.4), we conduct an ablation study by disabling the reference UV blending module (setting $\lambda$ to 0 in the experiment). As shown in Fig. 9, without the UV blending module, our method generates textures with noticeable distortions on the Joker's face over time.

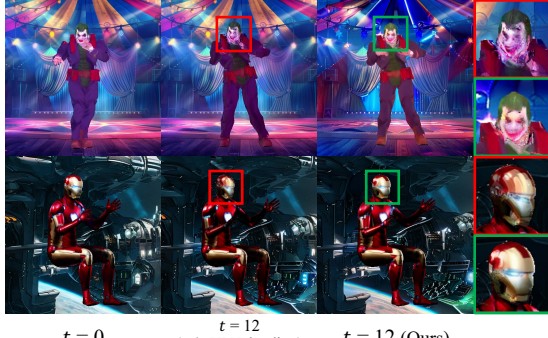

$t = 0$    $t = 12$ (w/o UV Blending)    $t = 12$ (Ours)

Figure 9. **Ablation study on the reference UV blending module.** Without this module, the generated textures lose consistency over time.

**Ablation for background priors.** Sec. 4.3 discusses the importance of including a background prior. To verify the effectiveness of this design, we replace the learnable background latents with an all-white background while keeping all other parts unchanged. Fig. 8 (c) shows that this ablation experiment produces significantly blurrier textures compared to our full method, highlighting the importance of background learning.

## 6 Conclusions

In this paper, we present Tex4D, a zero-shot approach that generates multi-view, multi-frame consistent dynamic textures for untextured, animated mesh sequences based on a text prompt. By incorporating texture aggregation in the UV space within the diffusion process of a conditional video diffusion model, we ensure both temporal and spatial coherence in the generated textures. To leverage priors from existing video diffusion models, we develop an effective modification to the DDIM sampling process to address the variance shift issue caused by multi-view texture aggregation and design a background learning module. Additionally, we enhance temporal consistency by introducing a reference UV map and developing a dynamic background learning framework to produce fully textured 4D scenes. Extensive experiments show that our method can synthesize realistic and consistent 4D textures, demonstrating its superiority against state-of-the-art baselines.

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

# Appendix

# A  Implementation Details

---

**Algorithm 1** Tex4D

---

**Input:** UV maps $\mathcal{UV} = \{UV_1, ..., UV_K\}$; depth maps $\mathcal{D} = \{D_{1,1}, ..., D_{1,V}, D_{2,1}, ..., D_{K,V}\}$; text prompt $\mathcal{P}$; CTRL-Adapter model $\mathcal{C}$; rendering operation $\mathcal{R}$; unproject operation $\mathcal{R}^{-1}$; cameras $\boldsymbol{c}$; $T$ diffusion steps; $\mathcal{T}$ latent textures (including foreground and background); $\lambda$ blending weight; $k$ keyframes

$\mathcal{T}_T \sim \mathcal{N}(\mathbf{0}, \mathcal{I})$       // Sample noise in UV space
$\tilde{\boldsymbol{z}}_T, \boldsymbol{\mathcal{M}}_{\text{fg}} = \mathcal{R}(\mathcal{T}_T; \boldsymbol{c})$
$\boldsymbol{z}_{b,T} \sim \mathcal{N}(\mathbf{0}, \mathcal{I})$
$\boldsymbol{z} = \boldsymbol{z}_T = \tilde{\boldsymbol{z}}_T \odot \boldsymbol{\mathcal{M}}_{\text{fg}} + \boldsymbol{z}_{b,T} \odot (1 - \boldsymbol{\mathcal{M}}_{\text{fg}})$       // Composite latents
**For** $t = T, \ldots, 1$ **do**
  $\boldsymbol{z}_{b,t-1} \leftarrow \mathcal{C}(\boldsymbol{z}_{b,t}; \mathcal{D}, \mathcal{P})$
  $\boldsymbol{\epsilon}_\theta \leftarrow \mathcal{C}(\boldsymbol{z}_t; \mathcal{D}, \mathcal{P})$       // Estimate noise from $\mathcal{C}$
  $\hat{\boldsymbol{z}}_0(\boldsymbol{z}_t) = \sqrt{\alpha_t} \cdot \boldsymbol{z}_t - \sqrt{1 - \alpha_t} \cdot \boldsymbol{\epsilon}_\theta$
  $\hat{\mathcal{T}}_0, \boldsymbol{\mathcal{M}}_{\mathcal{UV}} \leftarrow \mathcal{R}^{-1}(\hat{\boldsymbol{z}}_0; \boldsymbol{c}, \mathcal{UV})$       // Bake textures by Eq. 3
  $\mathcal{T}_{\mathcal{UV}} = \text{Combine}(\hat{\mathcal{T}}_0; \boldsymbol{\mathcal{M}}_{\mathcal{UV}})$
  **For** $k$ in $1, ..., K$ **do**
    $\mathcal{T}_{t-1}^k = \sqrt{\alpha_{t-1}} \cdot \hat{\mathcal{T}}_0^k + \sqrt{1 - \alpha_{t-1}} \left( \sqrt{\frac{\alpha_t}{1-\alpha_t}} \cdot (\sqrt{\alpha_t}\mathcal{T}_t^k - \hat{\mathcal{T}}_0^k) + \sqrt{1 - \alpha_t} \cdot \mathcal{T}_t^k \right)$    // Denoise Eq. 5
    $\mathcal{T}_{t-1}^k = \left((1 - \lambda) \cdot \mathcal{T}_{t-1}^k + \lambda \cdot \mathcal{T}_{\mathcal{UV}}\right) \odot \boldsymbol{\mathcal{M}}_{\mathcal{UV}}^k + \mathcal{T}_{\mathcal{UV}} \odot \left(1 - \boldsymbol{\mathcal{M}}_{\mathcal{UV}}^k\right)$    // Blend textures by Eq. 7
  $\tilde{\boldsymbol{z}}_{t-1}, \boldsymbol{\mathcal{M}}_{\text{fg}} = \mathcal{R}\left(\mathcal{T}_{t-1}; \boldsymbol{c}, \mathcal{UV}\right)$
  $\boldsymbol{z}_{t-1} = \tilde{\boldsymbol{z}}_{t-1} \odot \boldsymbol{\mathcal{M}}_{\text{fg}} + \boldsymbol{z}_{b,t-1} \odot (1 - \boldsymbol{\mathcal{M}}_{\text{fg}})$       // Composite latents by Eq. 6
  $\boldsymbol{z} = \boldsymbol{z}_{t-1}$
**Output:** $\boldsymbol{z}$

---

## A.1  Implementation Details

We utilize the CTRL-Adapter (Lin et al., 2024), trained on the video diffusion model I2VGen-XL (Zhang et al., 2023b), as the backbone for generation, with the denoising steps set to $T = 50$. Initially, we center the untextured mesh sequence and pre-define six different viewpoints around the Y-axis in the XZ-plane, uniformly sampled in spherical coordinates, along with an additional top view with an azimuth angle of zero and an elevation angle of 30°. For latent initialization, we first sample Gaussian noise on the latent textures and then render 2D latent samples for each view to improve the coherence of the generated outputs. During denoising, we upscale the latent resolution to $96 \times 96$ to reduce aliasing. We empirically set the blending coefficient to 0.2 during our experiment. It takes approximately 30 minutes to generate a video with 24 keyframes taken on an RTX A6000 Ada GPU. We decode the denoised latents in keyframes to RGB images, and then un-project and aggregate these images to transform the latent UV maps to RGB textures as previous works (Liu et al., 2023b; Cao et al., 2023; Huo et al., 2024). The resolutions of latent texture and RGB texture are $1024 \times 1024$ and $1536 \times 1536$, respectively. Finally, we interpolate the textures of the keyframes at intervals of 3 to synthesize the final video clips.

**Background Optimization** For each frame, we use a latent texture as the background shared across multiple views. The first frame is initialized from the image provided by the user as the CTRL-Adapter input. Then, we jointly optimize the background and foreground as described in Sec. 4.3. In our early attempts, we also used a cube or a hemicube mesh to model a free-view background, but found that the backgrounds were not accurate within uncaptured regions by cameras, such as the bottom. Note that our main focus is not a solution for free-view background generation, which is an open research problem explored by concurrent works like Cat4D (Wu et al., 2024) and GenXD (Zhao et al., 2024).

**Texture Interpolation Details** We compute RGB textures for non-key frames by interpolating RGB textures of nearby keyframes. Specifically, when the denoising stage finishes, we have RGB textures for

keyframes, i.e., $T_{rgb,t} = \frac{t-k\cdot n}{n} \cdot T_{rgb,k} + \frac{(k+1)\cdot n-t}{n} \cdot T_{rgb,k+1}$, where $t$ is the $t$-th frame in the final video, $k$ is the nearest prior keyframe, and $n$ is the keyframe interval.

**Ablation Algorithm Details** We provide the implementation details of our ablation algorithms as we used in Fig. 8. The algorithm (a) is similar to our full algorithm but overlooks the variance shift problem, and algorithm (b) utilizes texture domain aggregation for synchronized multi-view latents but does not denoise the noisy latents in the texture domain.

- Algorithm (a) cacluates multi-view $z_0$ $(z_{t,i})$ and aggregates them in the UV space, denoted as $\hat{T}_0$. Then aggregates multi-view $\epsilon_\theta$ $(z_t)$ in the UV space, denoted as $\epsilon_{\theta,t}$. Finally, it updates the latent texture $T_{t-1} = \sqrt{\alpha_{t-1}}\hat{T}_0 + \sqrt{1-\alpha_{t-1}}\epsilon_{\theta,t}$ by DDIM and then render the $z_{t-1}$ by $T_{t-1}$.

- Algorithm (b) first gets predicted multi-view $z_{t-1,i}$ in the previous step and then aggregates them to $T_{t-1}$ and finally renders $z_{t-1}$ from $T_{t-1}$ for synchronization.

**FVD Computation** We follow the prior paper SV4D (Xie et al., 2024) to compute the FVD metric. Specifically, we first render multi-view videos by sampling sequences of frames across different camera viewpoints. Then, for each generated video, we extract spatiotemporal features using the I3D network pretrained on Kinetics, as is standard in FVD. The FVD score is computed by fitting Gaussian distributions to the extracted features from generated videos and reference videos, and then measuring the Fréchet distance between the two distributions. For the measure of multi-view consistency, we evaluate FVD across all frame pairs, which reflects the semantic alignment and temporal coherence of multi-view sequences.

## A.2 Variance Shift Problem

In this section, we illustrate the vairance shift problem in detail. In the DDIM denoising process, the predicted noise $\epsilon_\theta$ is the estimated noise from the U-Net denoising module, which is expected to follow $\mathcal{N}(0,\mathcal{I})$. The variance of the latent $z_t$ at time $t$ is expected to be $\mathrm{Var}(z_t) = (1-\alpha_t)\mathcal{I}$ as proved in (Song et al., 2020) (We ignore the identity matrix $\mathcal{I}$ in the following for simplicity). However, the naive multi-view latents aggregation method (Eq. 3) would cause diminished distribution variance due to the weighted sum operation. Formally, for a given camera view $k$ at time $t$, we have the view latent $z_{t-1}^k$ that has at most $V_1$ latents overlapped in the UV space (including itself):

$$\begin{aligned}
\mathrm{Var}\left(\sum_{v=1}^{V_1} z_{t-1}^k\right) &= \mathrm{Var}\left(\sqrt{\alpha_{t-1}} \cdot \hat{z}_0(z_t^k)\right) + \mathrm{Var}\left(\sum_{v=1}^{V_1}\sqrt{1-\alpha_{t-1}}\epsilon_\theta(z_t^k) \cdot w_v\right) \\
&= 0 + \sum_{v=1}^{V_1}\mathrm{Var}\left(\epsilon_\theta(z_t)\right) \cdot (1-\alpha_{t-1}) \cdot w_v^2 \\
&< (1-\alpha_{t-1}) \cdot \mathcal{I} \cdot (\sum_{v=1}^{V_1} w_v)^2 \\
&\leq (1-\alpha_{t-1}) \cdot \mathcal{I}
\end{aligned} \tag{8}$$

where $w_v = \cos{(\theta^v)}^\alpha / \sum_{v=1}^{V}\cos{(\theta^v)}^\alpha$ is a constant related to predefined cameras and the baking power $\alpha$. Note that we have $\sum_{v=1}^{V} w_v \leq 1$. The variance shift problem stems from the aggregation operation (weighted sum).

We observe that this issue could be resolved by rewriting $\epsilon_\theta(z_t^k)$ with the combination of $\hat{z}_0^k$ and $z_t^k$, and denoising in the UV space (Liu et al., 2023b). First, the multi-view aggregation of $\{\hat{z}_0^k\}_{k=1}^V$ (i.e., $\hat{\mathcal{T}}_0$) has variance $= 0$, as each component in Eq. 3 is expected to be 0. Second, performing the rewritten DDIM denosing process (Eq. 5) in the UV space forwards $\mathcal{T}_{t-1}$ with $\mathcal{T}_t$, avoiding the aggregation of latent at timestep $t$ ($\{z_t^k\}_{k=1}^V$, which results in variance $\neq 0$). These two combinations effectively avoid the variance shift problem.

### A.3 Denoising Algorithm of Our Method

We present the complete workflow of our method in Algorithm 1. For clarity, we omit the notation for the latent variables $z_b$ representing the background plane texture, as they follow the same scheme as the foreground latents. The reference UV map $\mathcal{T}_{\mathcal{UV}}$ is constructed by progressively combining latent textures over time, with each new texture filling only the unoccupied texels in the reference UV map. We denote this process as "Combine" in the following workflow.

### A.4 Evaluation Details

**Details of Test Samples** For FVD and consistency score evaluation, our test set contains 20 distinct mesh sequences, including 7 non-human characters (from Sketchfab) and 13 human-like characters (from Mixamo and human diffusion models). This balance ensures diversity across both geometry and character type. To increase variation, we assign multiple stylization prompts per sequence, resulting in a total of 32 test cases. This setup allows evaluation of both generalization across diverse meshes and controllability under varied textual instructions.

**Details of User Study** We conducted a user study comparing 6 methods (Tab. 1, including ours) across 8 mesh sequences, producing 48 multi-view video results. The study followed an A/B test protocol (Fig. 18): participants viewed paired results and (1) selected the preferred method, and (2) rated each on a 1–5 scale for spatial and temporal coherence. For fairness, each video was composed of two randomly selected views from 8 predefined camera viewpoints.

#### A.4.1 Details of Baseline Methods

Because dynamic textures are mainly applied in dynamic video creation, we compare Tex4D against three categories of baselines: (i) zero-shot video generation, (ii) zero-shot texture generation, and (iii) 4D generation. Below, we describe the settings and limitations of each group.

**Zero-shot video generation methods.** PnP-Diffusion is designed for image-to-image stylization; we adapt it to video following TokenFlow: each frame of the clay-rendered mesh sequence is independently stylized. This straightforward extension provides intuitive zero-shot stylization but lacks temporal constraints, resulting in strong flickering artifacts across frames (see supplementary video).

TokenFlow, Text2Video-Zero, and LatentMan are models that perform video editing by directly altering the latent codes of diffusion models. In our evaluation, we first render an untextured animation video, then apply each model with textual prompts. While they generate plausible frame-wise stylizations, the view-sensitive nature of diffusion latents leads to inconsistent textures across different camera viewpoints, breaking multi-view coherence.

To ensure comprehensive coverage, we also include Gen-1, a commercial model that supports untextured video stylization. Following the TokenFlow protocol, we apply it to our clay-rendered animations. Although it often produces higher-quality stylization compared to open-source models, it still exhibits temporal flickering and lacks explicit mechanisms for multi-view consistency.

**Zero-shot texture generation methods** Dynamic textures offer two clear advantages over static ones: (1) they naturally represent motion patterns, and (2) they reduce untextured regions caused by occlusions, since areas hidden in one frame may appear in others. We evaluate Text2Tex (see Fig. 7) as a representative texture baseline. However, because it inpaints textures only in the canonical space, it frequently produces empty or incomplete textures, especially in regions not visible from the canonical viewpoint.

**4D generation methods** For completeness, we also compare against 4D generation approaches. These methods take an untextured mesh and textual prompt as input, similar to our pipeline. However, they typically rely on generative rendering pipelines instead of explicit UV-based modeling, which makes it difficult for them to maintain multi-view consistency across frames.

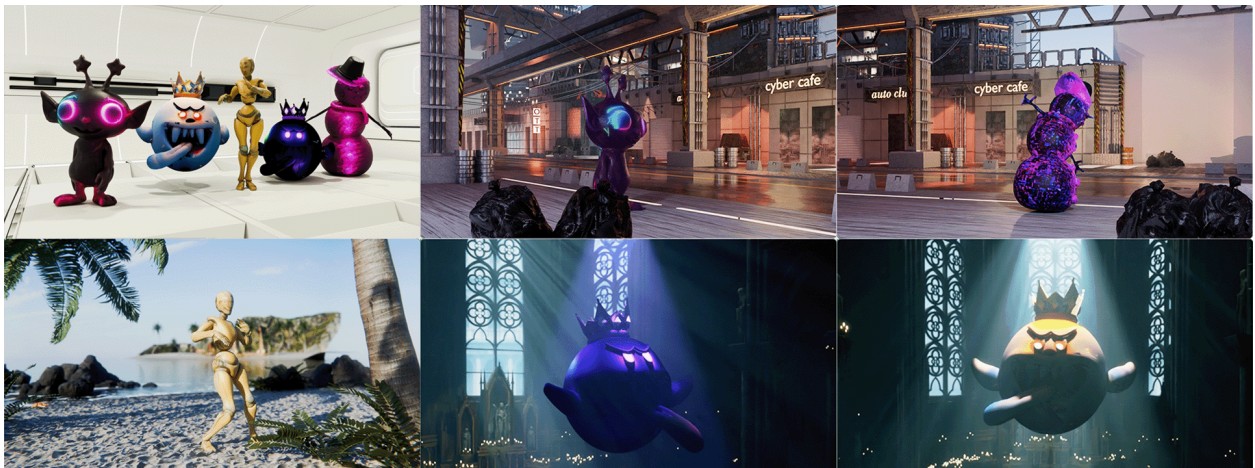

Figure 10. **Tex4D Applications.** Our synthesized dynamic textures can be easily integrated into graphics pipelines. We utilize the shader editor in Blender to animate textures with image sequence nodes. The dynamic textures help technical artists render vivid videos without additional lighting and mesh controls. We provide a video demo to further demonstrate the effectiveness of Tex4D.

## B  Additional Qualitative Results

In this section, we provide more qualitative results of Tex4D to deliver a comprehensive application and analysis of our method. First, we demonstrate the novel application of our method in collaboration with technical artists as in Sec. B.1. Our synthesized dynamic texture sequence can be seamlessly integrated into Blender's shader editor and we provide the rendered video demo in the supplementary material. Then, we analyze the multi-view rendering results of Tex4D with a diverse set of prompts and styles in Sec. B.2. Finally, we visualize the generated texture sequence from three different cases to demonstrate that our method can directly bake lighting variations and appearances into textures without the need for post-processing by technical artists in creating vivid videos.

### B.1  Graphics Application and Video Demo

As shown in Fig. 10, Tex4D demonstrates its utility in the graphics pipeline by integrating dynamic texture sequences into Blender for rendering. This integration enables seamless visualization of animated textures directly on 3D models, showcasing Tex4D's capability to handle complex visual dynamics in real-world applications. We highly recommend the reviewers watch our supplementary videos for details.

### B.2  Multi-view Results

In this section, we discuss the multi-view results of our methods and other methods. As shown in Fig. 13, our method generates vivid videos that align with the textual prompts while preserving spatial consistency. PnP-Diffusion (Tumanyan et al., 2023), TokenFlow (Geyer et al., 2023) and Text2Video-Zero (Khachatryan et al., 2023) generate videos that are not aligned with the text prompt, due to the implicit correspondence used in the multi-frame attention. LatentMan (Eldesokey & Wonka, 2024) and Gen-1 (Esser et al., 2023) generate vivid videos, but the multi-view consistency of the characters is not well-preserved.

In Fig. 11, we present additional characters generated by Tex4D, showcasing the method's effectiveness and its ability to generalize across a diverse array of styles and prompts. We also evaluate Tex4D on non-human character animations in Fig. 12, demonstrating its robust generalization capabilities across various types of mesh sequences. In each case, we provide two different views to show that our method can ensure multi-view consistency.

To emphasize the temporal changes in the generated textures, we also design some prompts, for example, 'flashed a magical light', 'dramatic shifts in lighting', and 'cyberpunk style' in our experiments. As shown in

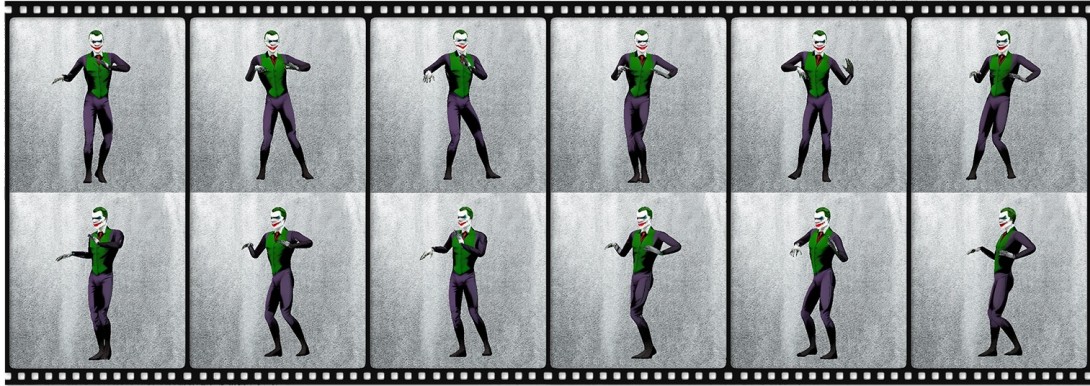

"the Joker dances, comic style"

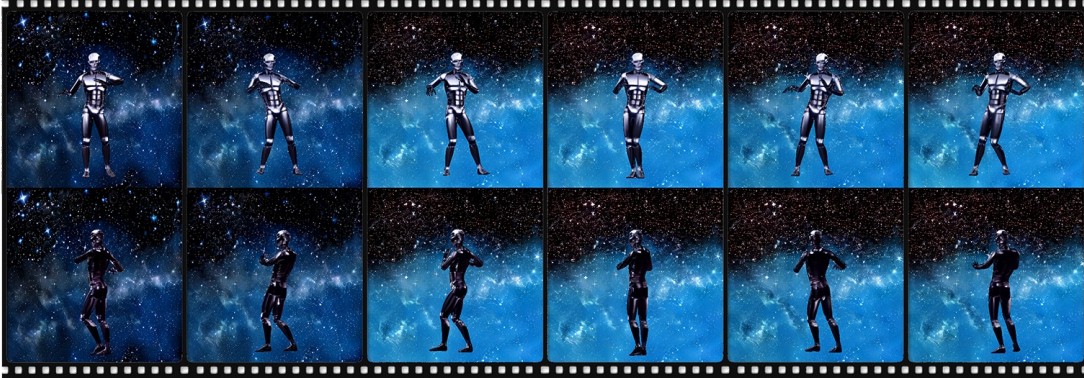

"the terminator dancing in the milky way"

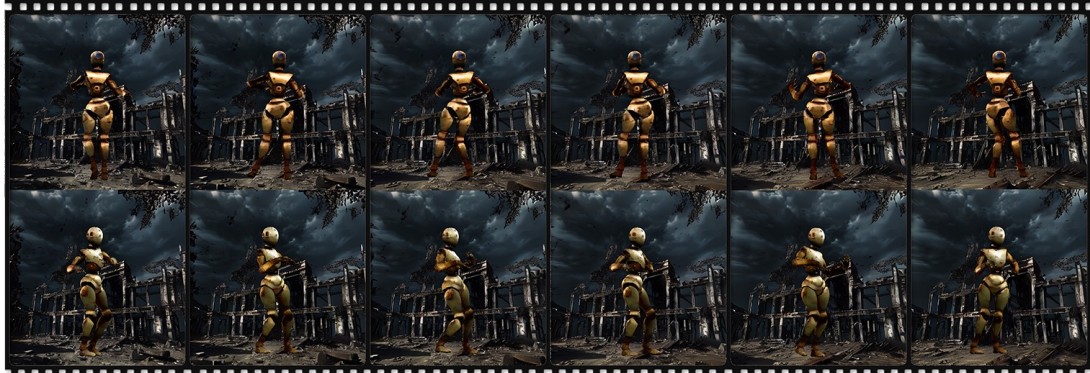

"a rusty robot dances in ruins"

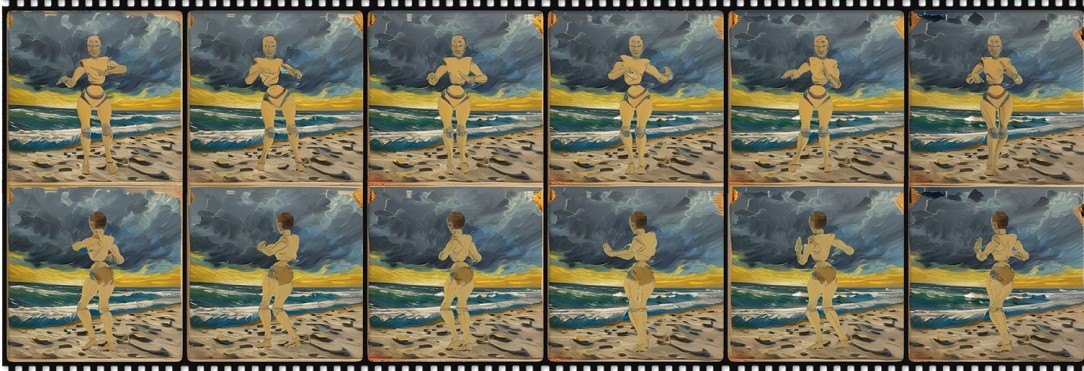

"a sketch of bot dancing in a sandy beach, Van-Gogh style."

Figure 11. **More qualitative results.** We present the results of Tex4D with brief prompts, demonstrating the ability of Tex4D to generate multi-view consistent textures.

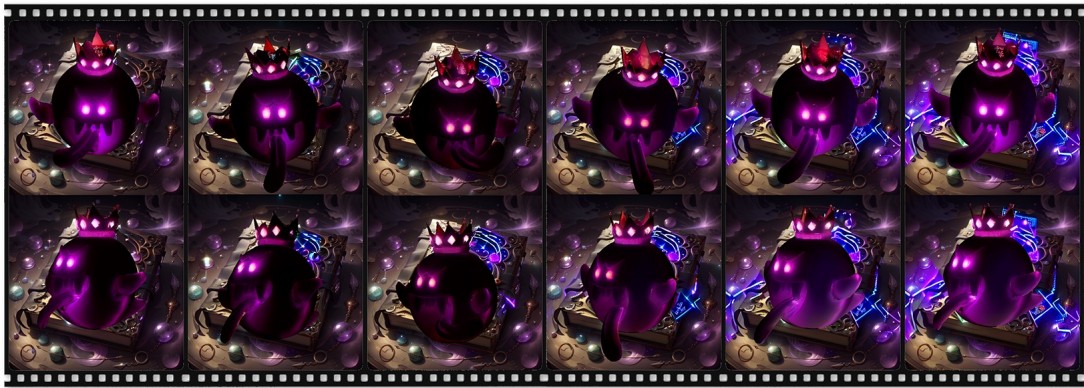

"a ghost flashed a magical light, causing dramatic shifts in lighting."

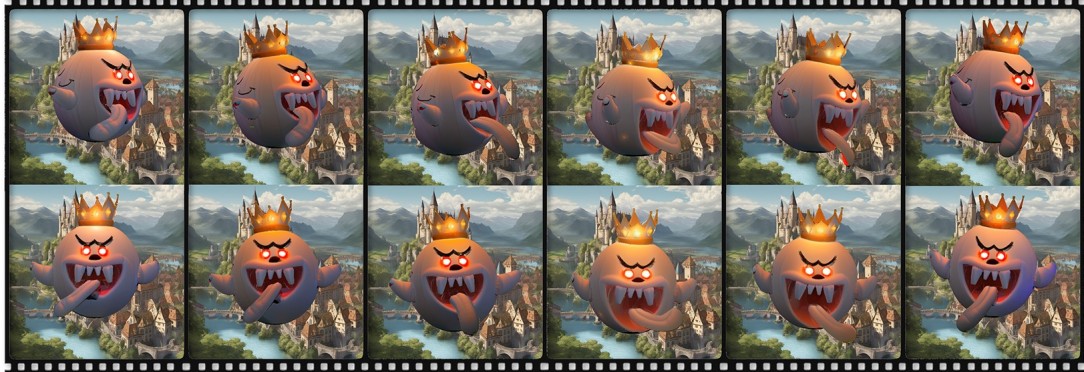

"a dingy, magic King Boo, flashing a weird light, static background."

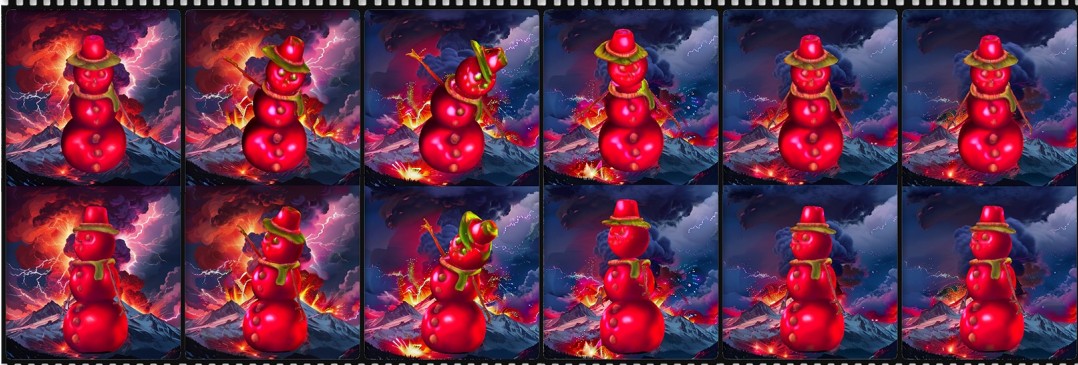

"a sprite of fiery plums tilts its head, in full color."

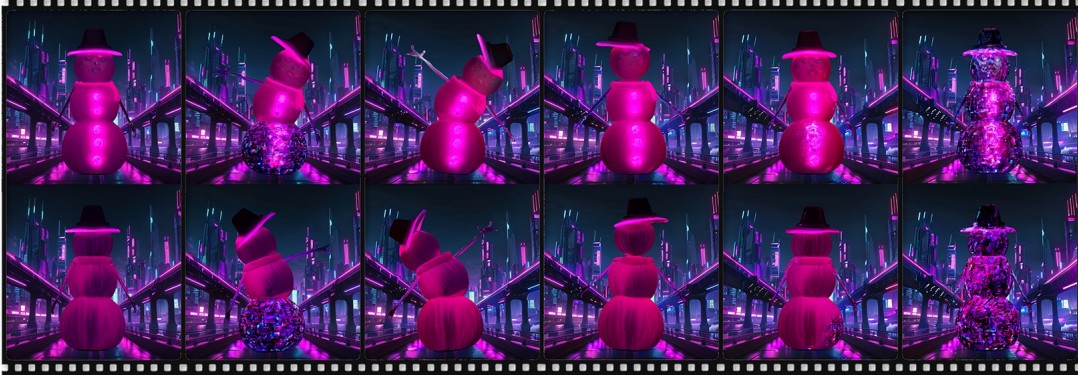

"a spirit in neon tilts its head, cyberpunk style."

Figure 12. **More qualitative results on non-human character animations.** We present the results of Tex4D with prompts emphasizing the dynamics, demonstrating the ability of Tex4D to capture the dynamics from video diffusion models.

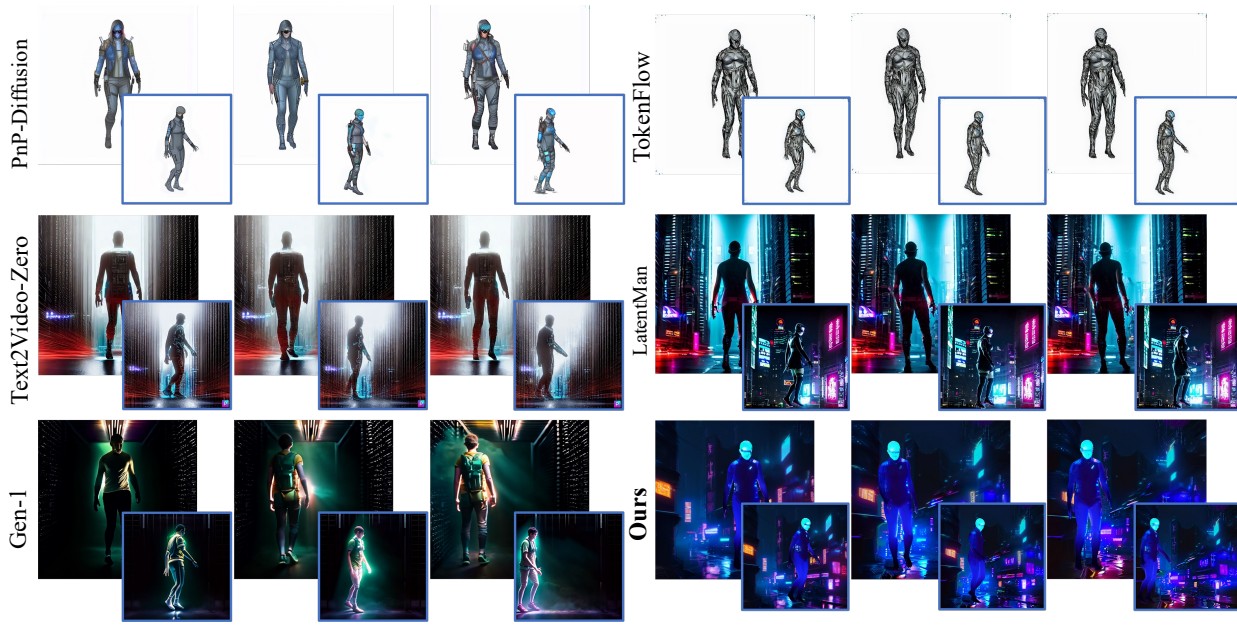

Figure 13. **Qualitative comparisons of multi-view video generation.** We compare our method against PnP-diffusion (Tumanyan et al., 2023), TokenFlow (Geyer et al., 2023), Text2Video-Zero (Khachatryan et al., 2023), Generative Rendering (Eldesokey & Wonka, 2024), and Gen-1 (Esser et al., 2023). We generate videos in the front view and the side view (blue box) on Human Diffusion Models. Our method generates vivid videos that align with the textual prompts while preserving spatial consistency.

Fig. 12, the results of 'ghost', 'King Boo' and 'Snowman' validate the effectiveness of our method in generating different level of temporal changes by a variety of textual prompts, while maintaining the consistency both spatially and temporally. Additionally, we provide a supplementary video that includes baseline comparisons and multi-view results for all examples.

### B.3 Texture Results

In this section, we present the texture sequences, which are the intermediate results of our pipeline. Our method utilizes XATLAS to unwrap the UV maps from meshes without human labor. XATLAS is a widely used library for mesh parameterization and UV unwrapping, commonly integrated into popular tools and engines, facilitating efficient texture mapping in 3D graphics applications. As shown in Fig. 14, our method seamlessly bakes temporal changes, including lighting variations, wrinkles, and appearance transformations, directly into the textures, removing the need for manual post-production by technical artists.

## C Additional Ablation Results

**Ablation on Background** To show the effects of various background latent initialization strategies, we provide additional examples, including the approach used in the texture synthesis method (Liu et al., 2023b) and a background that contrasts sharply with the foreground object, as shown in Fig. 15. In detail, SyncMVD (Liu et al., 2023a) encodes the backgrounds with alternative random solid color images. For the high-contrast background experiment, we use the latents obtained from the DDIM inversion of highly contrasted foreground and background to initialize the latents.

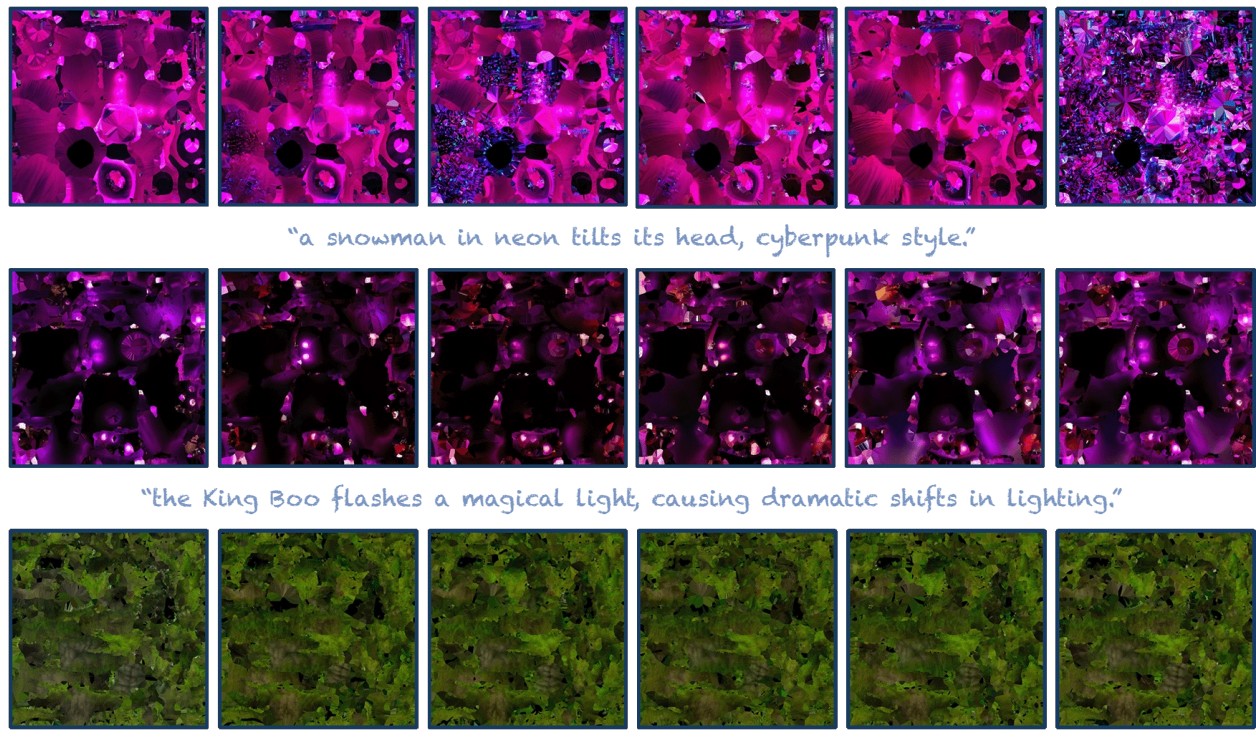

Figure 14. **Visualization of generated textures for mesh sequences.** Our method effectively incorporates temporal changes, such as lighting variations and appearance transformations, directly into the textures, eliminating the need for post-production by technical artists.

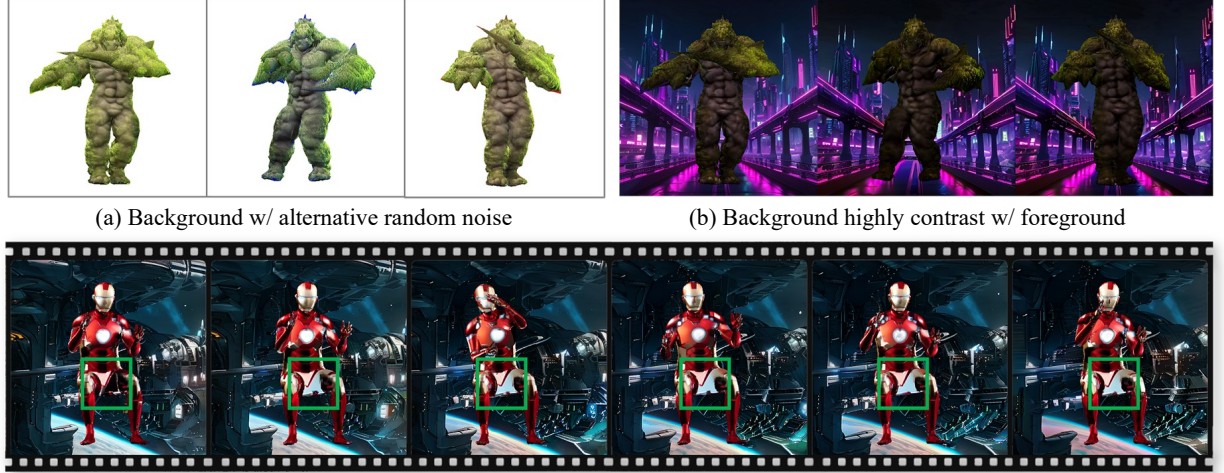

(a) Background w/ alternative random noise      (b) Background highly contrast w/ foreground

(c) Ours w/o background priors (foreground composited with our background)

Figure 15. **More ablation study on the background priors.** We present three ablations, including the approach used in the texture synthesis method SyncMVD (Liu et al., 2023b), a background that contrasts sharply with the foreground, and without background priors.

## D  Additional Performance Evaluations

### D.1  Comparison with Depth-Conditioned Video Diffusion Models

While depth-conditioned video diffusion models effectively generate visually compelling results from a single viewpoint, they often struggle to maintain consistent multi-view representations of a single object, such as a character, across different perspectives. To highlight this limitation, we present multi-view results from the

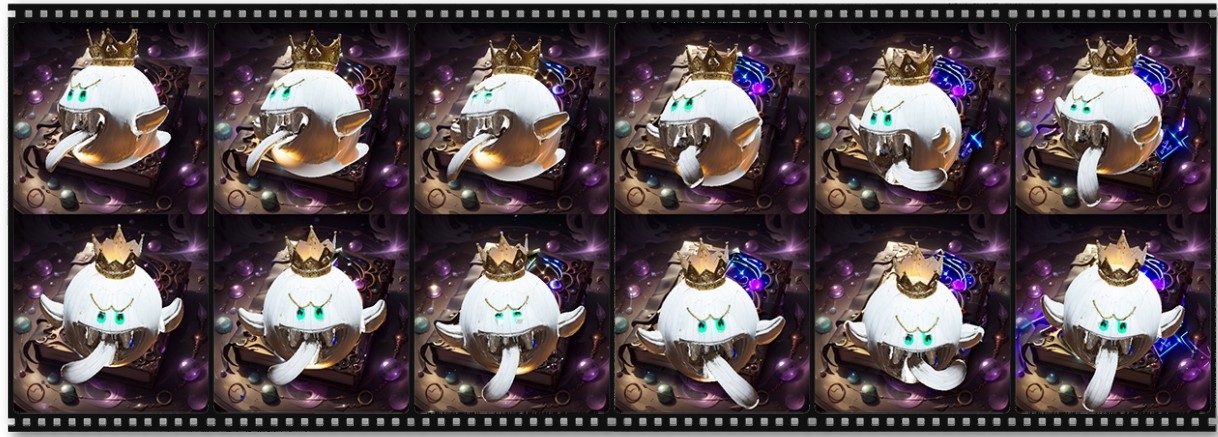

"the King Boo flashes a magical light, causing dramatic shifts in lighting."

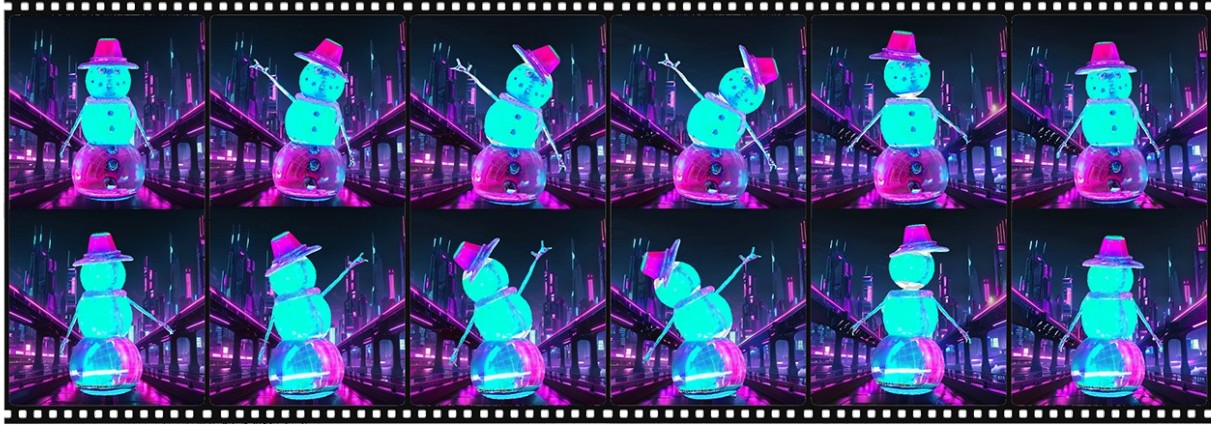

"a spirit in neon tilts its head, cyberpunk style."

Figure 16. **Results of textured mesh animation.** We present the visual results of Text2Tex (Chen et al., 2023b) with our backgrounds. Text2Tex fails to capture temporal variations between frames and results in empty texels in invisible regions.

depth-conditioned video diffusion model in Fig. 17. The primary cause of this issue is that depth conditions are inherently view-dependent and lack global geometry information, in contrast to UV maps, which provide global information about the 3D space, enabling a unique mapping for each 3D point across all views.

## D.2  Comparison with Textured Mesh Animations

In this section, we highlight the differences between our method and traditional approaches, demonstrating the effectiveness of 4D texturing in capturing temporal variations (e.g., lighting and appearance transformation) within mesh sequences to produce vivid visual results. Traditional methods typically involve texturing a base mesh (often called a clay mesh) and animating it using skinning techniques. This approach lacks the capacity to represent dynamic visual transformations, and the animated sequence is then refined by technical artists who control scene lighting or simulate cloth dynamics to achieve the final visual presentation. This process is labor-intensive and demands specialized expertise in filmmaking and technical engines.

Instead, Tex4D introduces the first solution to dynamic texture generation by leveraging the expressiveness of video diffusion models. This is a fundamental task in filmmaking and game design to model character appearance transformations.

Compared with the static texture generation method, our method presents an alternative by directly integrating complex temporal changes into mesh sequences. As shown in Fig. 4, 11 and 12, our approach

effectively captures temporal effects such as dynamic lighting, and evolving appearances using textual prompts, significantly simplifying the workflow while maintaining high-quality results.

We demonstrate the limitations of traditional textured mesh animation in handling complex temporal changes in Fig. 16. Specifically, we employ the Text2Tex (Chen et al., 2023b) to generate the texture for the input mesh in T-pose and render it from multiple viewpoints. To ensure a fair comparison, we composite the rendered results with the background generated by our method. Notably, the 'ghost' and 'snowman' examples exhibit visible seams during animation due to self-occlusions that commonly appear in dynamic poses but cannot be accurately predicted during T-pose texture generation. Although the texture is still globally consistent, Text2Tex not just fails to model dynamic effects like appearance transformation but also results in empty texels and disrupts the visual continuity of the animation. For the rendered video results, please kindly refer to our supplementary videos.

### D.3 Comparison with Text-to-4D Methods

Although the setting of the Text-to-4D task is different from Tex4D as the mesh sequence is not given, we also provide a comprehensive comparison for Text-to-4D methods like SV4D Xie et al. (2024) and L4GM Ren et al. (2024) as in Fig. 5. In our experiments, we found that these Text-to-4D methods usually fail to generate plausible results and have these limitations:

- Multiview video-based methods (e.g., SV4D) struggle with consistency under significant motion. In our early attempts, we also found that the diversity of multiview attention-based inference heavily depends on dataset quality. SV4D handles only simple character animations.
- Animatable Gaussian-based methods (DreamGaussian4D, L4GM) built on LGM suffer from blurry and static textures, as decoupling geometry and appearance simultaneously is more challenging.

## E   User Study

Our study included 67 participants who provided 1,104 valid responses across six scenes from previous works, each rendered from two different viewpoints. We show each participant 30 pairs of videos synthesized by different methods, capturing the same object from different views. For each pair, each participant is asked three questions in sequence:

- Which method has better appearance quality?
- Which method has better spatial and temporal consistency?
- Which method has better fidelity to the prompts?

Our study involved 67 participants who provided a total of 1104 valid responses based on six different scenes drawn from six previous works, with each scene producing videos from two different views.

In addition, we further invite 24 participants who provided 455 valid trials to statistically evaluate user preferences (rating from 1 to 5) across Appearance quality, Spatio-temporal consistency, and Prompt Consistency. Users consistently favored Tex4D over all baselines.

## F   Broader Discussion

As Tex4D is the first work to generate dynamic textures, we provide a broader analysis of our method in this section and hope to provide more insights for future works.

### F.1   Robustness of Different UV Mappings

To validate the robustness of our method in different UV mappings, we test Tex4D using Blender's smart UV strategy. Specifically, we set the angle limit for adjacent face normals to 30 degrees for splitting with more seams. The rendered results are also visually robust as shown in Fig 19.

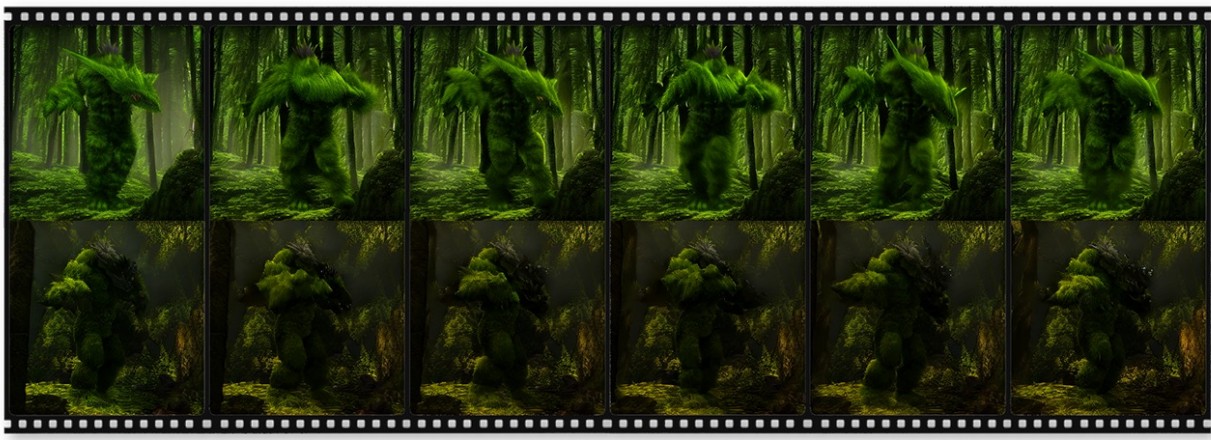

"a mossy stone monster dances in a mysterious forest."

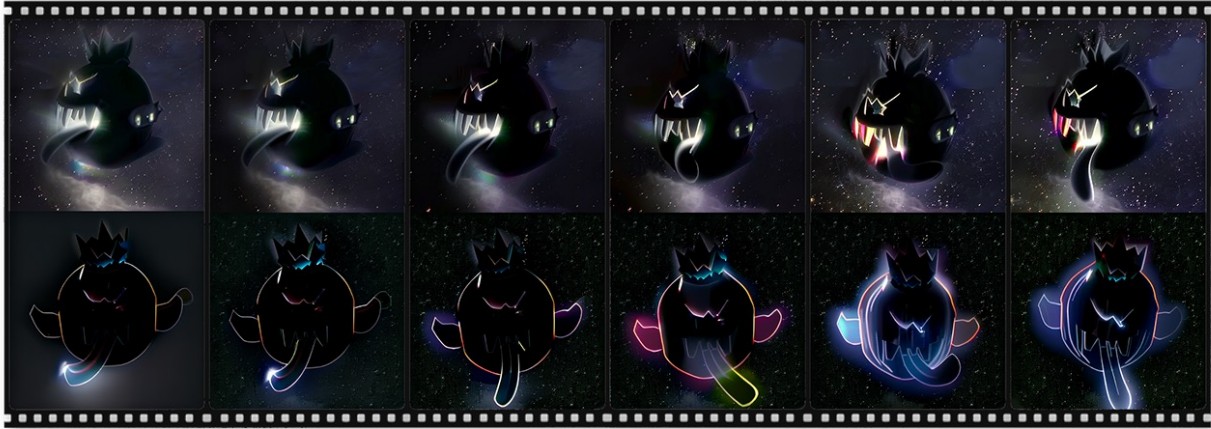

"a ghost flashed a magical light, causing dramatic shifts in lighting."

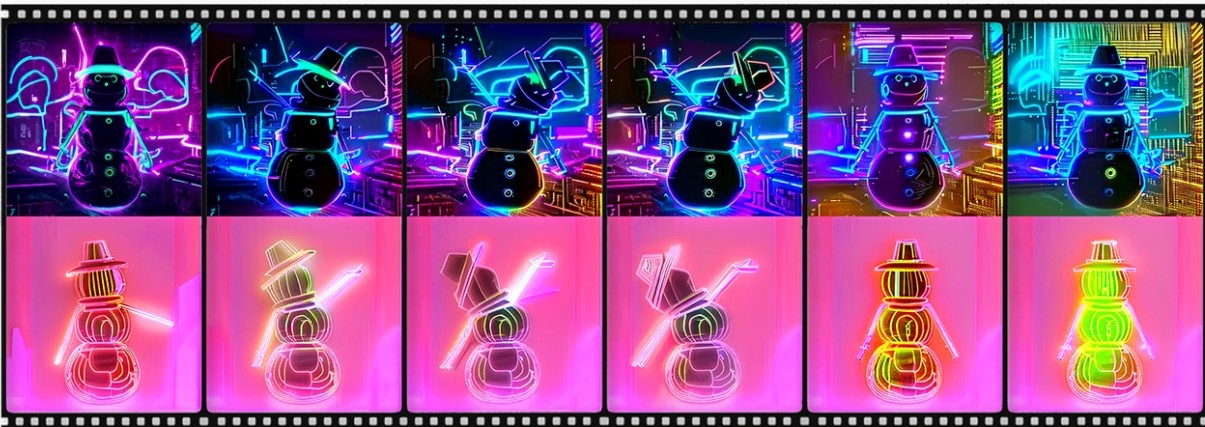

"a spirit in neon tilts its head, cyberpunk style."

Figure 17. **Multi-view results from conditioned video diffusion models.** The conditioned video diffusion models struggle to maintain consistent multi-view representations of a single object due to the depth condition being view-dependent and lacking global geometry information.

## F.2 Highly Structured Texture Generation

To further demonstrate the robustness of our method, we test Tex4D with the task of highly structured texture generation that is commonly used in relief mapping and displacement mapping. We create a low-poly house as the base mesh and test our method with different prompts as shown in Fig. 20. Tex4D could

Please find the method that has **best spatial and temporal consistency.** The prompt is "a Stormtrooper swimming"
Videos are capturing the same object from different views.

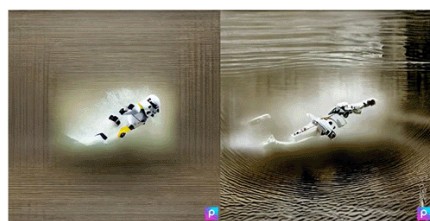 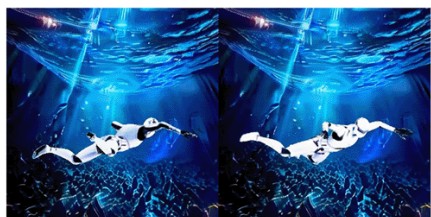

Next

3% (1 / 30)

Please find the method that has **best spatial and temporal consistency.** The prompt is "a cyberpunk walks"
Videos are capturing the same object from different views.

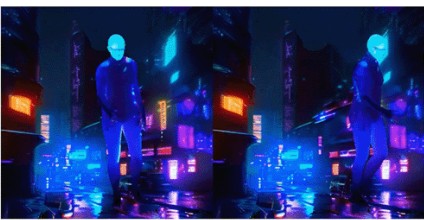 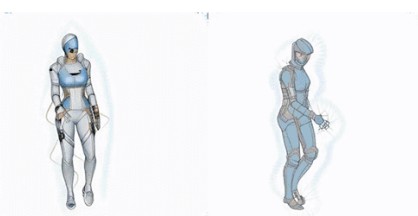

Next

13% (4 / 30)

Please find the method that has **best fidelity to the prompt.** The prompt is "Ironman turns steering wheel"
Videos are capturing the same object from different views.

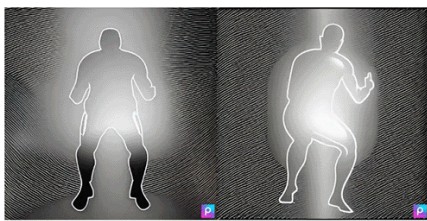 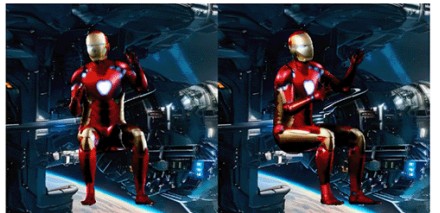

Next

96% (29 / 30)

Figure 18. **User Study.** We provide more visual examples and include quantitative results from our user study. We evaluate the videos from three metrics: Appearance Quality, Spatial and Temporal Consistency, and Fidelity to the Prompt.

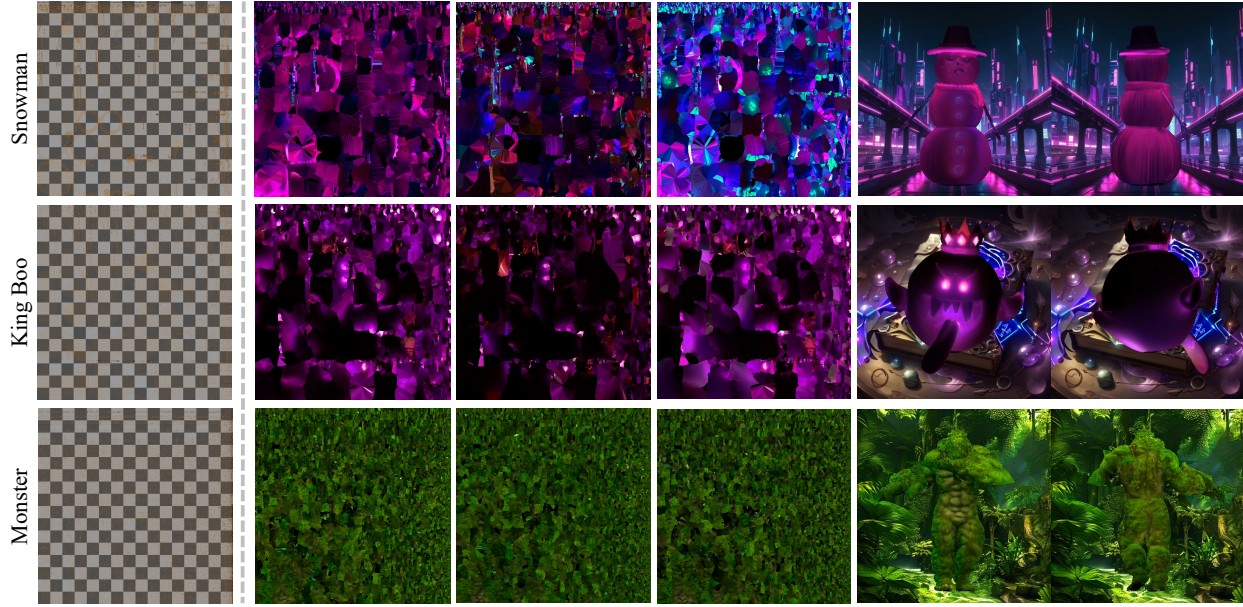

Figure 19. Complex UV with seams created by Blender Smart UV (angle limit=0.5). Tex4D is robust with different UV mappings. First column: UV without texels. Last column: rendered video frames. Other columns: texture generated by Tex4D. Zoom in for texture details.

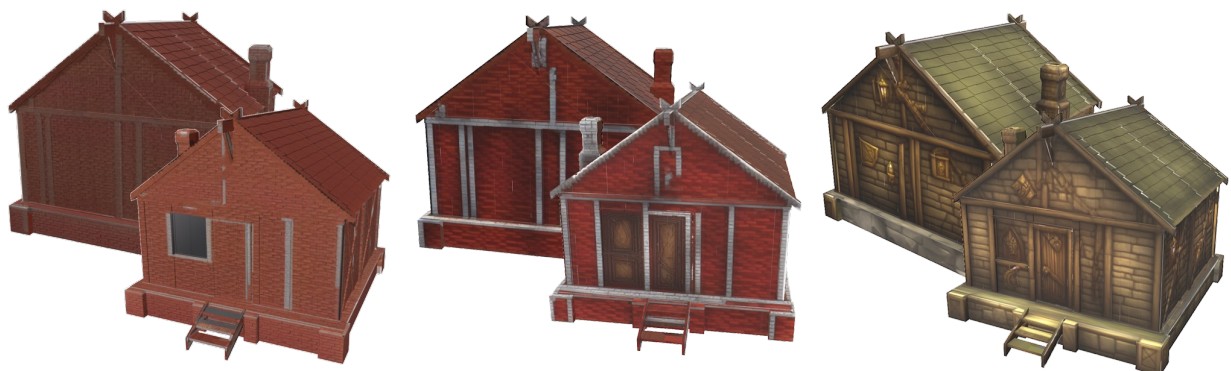

Figure 20. **Highly Structured Texture Generation.** A low-poly house is created with different prompts to demonstrate the capability of Tex4D to generate diverse, highly structured textures. Zoom in for texture details.

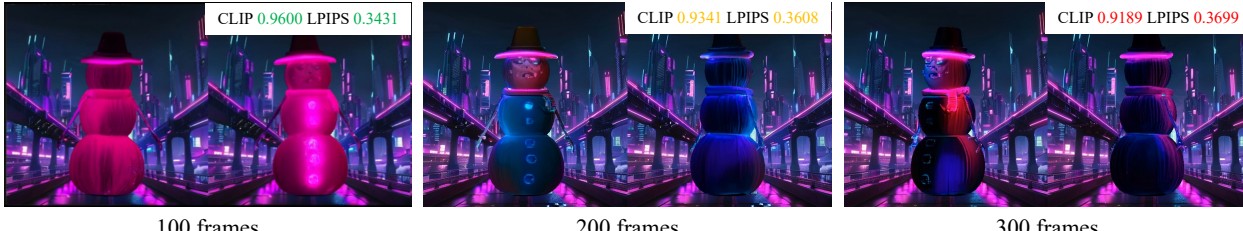

CLIP 0.9600 LPIPS 0.3431     CLIP 0.9341 LPIPS 0.3608     CLIP 0.9189 LPIPS 0.3699

100 frames        200 frames        300 frames

Figure 21. **Long Texture Sequence Generation.** Our method could maintain plausible textures with the increase in animation sequences. However, the quality may degrade due to the training sets of video diffusion models usually do not contain such long sequences.

generate multi-view consistent and highly structured texture, demonstrating our method is robust across different tasks.

### F.3 Runtime Breakdown

In Tab. 2, we provide a runtime breakdown including different keyframe numbers and latent resolutions. We have tested the latent resolutions with $96 \times 96$ and $64 \times 64$, and the keyframe numbers 3, 10, 17 and 25 respectively in the video generation task with a total frame number of 72. In our default setting, we use the keyframe interval of 3 and latent size $96 \times 96$ for better generation quality.

| Keyframes | 3 | 10 | 17 | 25 |
|---|---|---|---|---|
| Time (min) w/ res. (96×96) | 4.1 | 13.6 | 26.6 | 34.4 |
| Time (min) w/ res. (64×64) | 2.3 | 6.8 | 10.7 | 14.4 |

Table 2. **Quantitative results of runtime breakdown.**

### F.4 Long Texture Sequence Generation

In this section, we extend the original mesh sequence to longer lengths to study the robustness of Tex4D at long texture sequence generation. Specifically, we use the "snowman" case, which originally has 100 frames. We extend the mesh sequence to 200 and 300 frames by repeating the animation. Our experiments show that Tex4D remains robust when generating these longer sequences. The method successfully preserves both the temporal dynamics of the animation and the overall appearance consistency of the textured surfaces. Representative results of intermediate frames are shown in Fig. 21, where the textures maintain coherence despite the extended length.

However, we also observe gradual degradation in texture detail as the sequence length increases. Specifically, high-frequency components such as fine surface patterns or sharp boundaries become less stable, leading to subtle distortions. We report the CLIP (Radford et al., 2021) and LPIPS (Zhang et al., 2018) scores between the first and 100th frames in the top-right region. The results show that as the number of generated frames increases, alignment with the initial frame tends to degrade. This phenomenon is consistent with prior findings on video diffusion models: Diffusion models tend to smooth out high-frequency details over long horizons, which causes a loss of sharpness and fidelity in extended outputs (Lu et al., 2024). Furthermore, existing video diffusion training sets typically contain only short to medium-length sequences, making it difficult for models to generalize to long-duration appearance modeling (Chen et al., 2024). As we indicated in Fig. 21, the high-frequency details of snowman will disappear, which aligns with the observations in the video diffusion models.

### F.5 Analysis of Appearance Details

During our early experiments and literature review, we found video diffusion models tend to generate relatively smooth results compared with image diffusion models, as indicated by DiffusionRenderer Liang et al. (2025) and I2VGen-XL (Zhang et al., 2023b). As a result, in some cases, the clown's face lacks fine details within small pixel patches as shown in Fig. 9.

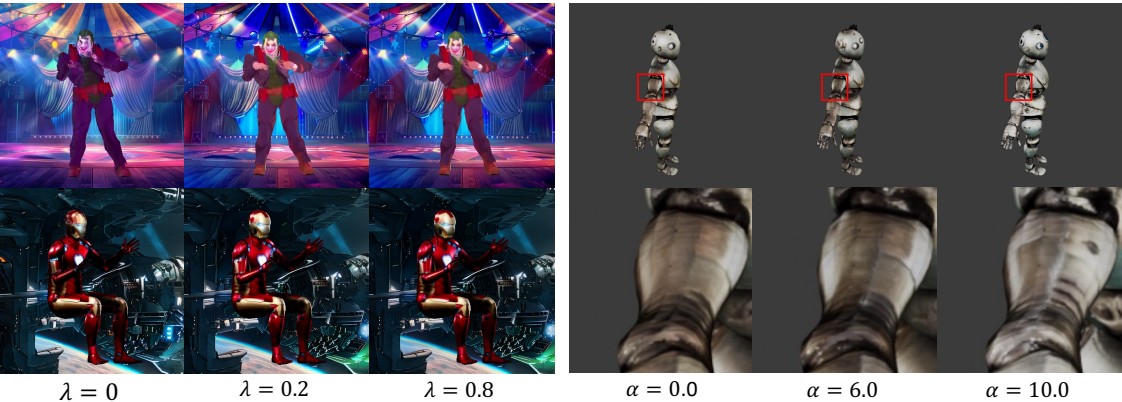

Figure 22. Effects on Hyperparameters. We study the effects of hyperparameters, the UV blending coefficient $\lambda$ and the aggregation power $\alpha$. Our method is generally robust under different values. However, an excessive $\alpha$ value would cause visible gaps in the texture, as the texture projection process is dominated by the cosine value of view angles.

We analyze the underlying causes from three perspectives. Intuitively, maintaining high-frequency details temporally is harder than a smooth one, and video diffusion models tend to comprise the spatial details for better temporal consistency. Technically, this problem should be attributed to the data limitations, including dataset scales and low-quality patches from websites (e.g., WebVid10M (Bain et al., 2021)) used for video diffusion models, as analyzed by VideoCrafter2 (Chen et al., 2024). Specifically for Tex4D, the latents aggregation module (Sec. 4.2) may potentially wipe out details because of the average operation on latents, although with the weighted cosine maps. A possible step is to train a tiny adapter like TexFusion Cao et al. (2023) to optimize the latents aggregation instead of RGB mapping.

Some concurrent works attempt to address this problem, either from the data distribution view (FreqPrior (Yuan et al., 2025)) or the U-Net architecture view (CogVideoX (Yang et al., 2024)). As our methods can be easily integrated into any video diffusion model, we anticipate improving the quality of Tex4D with the advancement of video diffusion models.

## F.6 Hyperparameters

We evaluate two key hyperparameters in our design: the UV blending coefficient $\lambda$ and the aggregation power $\alpha$. As shown in Fig. 22, our method remains generally robust to variations in $\lambda$, whereas an excessively large $\alpha$ can introduce visible gaps in the texture.

## F.7 V-Prediction

Tex4D is a zero-shot approach built on a pre-trained conditional video diffusion model, where v-prediction is a technique commonly used in video diffusion models (e.g., I2VGen-XL (Zhang et al., 2023b), Imagen (Ho et al., 2022), CogVideoX (Hong et al., 2022; Yang et al., 2024), CogView3 (Zheng et al., 2024)) to accelerate the training and prevent temporal color shifts. In our method, we utilize CTRL-Adapter (Lin et al., 2024), a conditional video diffusion model that guides video by depth maps trained on the DDIM v-prediction mechanism. Hence, we use v-prediction to ensure the proper functioning of the conditional video diffusion model.

## F.8 Societal Impacts

**Potential Social Impacts** Although our method offers broad potential for controllable generation tasks, like other frameworks for image and video synthesis, it could be misused for harmful purposes (such as generating deceptive content or fake media). As such, responsible and cautious use is essential when applying it in real-world scenarios.

**Safeguards** During inference, we enable the NSFW filter provided by the underlying models to help prevent the generation of explicit or inappropriate content, thereby safeguarding users from unwanted exposure. The training datasets used in our base model, CTRL-Adapter, have already filter out the image/video samples with harmful contents. Specifically, Panda70M filters harmful content using an automated pipeline and replaces names with "person" via NLTK. Similarly, LAION-POP applies a custom NSFW classifier to exclude unsafe samples.

## G  Limitations

### G.1  Panoramic Background Modeling

One limitation of our method is the lack of seamless integration between the generated textures and the background, resulting in a disjointed appearance where the foreground and background elements may seem artificially stitched together. However, the dynamic textures remain globally consistent and can be directly applied to the downstream tasks, as shown in Fig. 10. To the best of our knowledge, no existing work tackles the foreground and background texture generation together because the task is computationally expensive, and the scene-level dataset is limited. Addressing the scene-level 4D texturing remains an open challenge that we aim to explore in future work.

### G.2  Computation Time

We notice that our method is relatively computationally intensive compared with other texture synthesis methods. The running time of our method primarily depends on the foundation model CTRL-Adapter, taking 5 minutes to generate a 24-frame video. We anticipate efficiency improvements with advancements in conditioned video diffusion models to further enhance the practicality of Tex4D.

