# OpenReview forum: "Tex4D: Zero-shot 4D Character Texturing with Video Diffusion Models"
_TMLR — Rejected by TMLR_

### Review · Reviewer_Xpqh · 2025-08-25

**Summary Of Contributions:**

This paper presents a mesh-based video generation pipeline, using video diffusion models as the main engine together with some modifications that seem to improve the video quality in three different metrics (Fréchet Video Distance, Consistency Score, and human preference) when compared with baselines. The main contribution is the modifications done in the diffusion process: meshes from different views are aggregated in the UV space to get a better alignment, together with some patches to further improve that.

**Additional Comments:**

These are minor comments.

One general piece of advice is to rephrase parts that are obvious to people who work in this subdomain but not so much to others, to increase readability and broaden impact. E.g., imagine as if a fellow RL researcher is going to read this paper. This doesn’t have to be lengthy explanations but small hints here and there. Including but not limited to,
- What is the UV space?
- What is DDIM?
- What’s a T2I model, text-to-image?
- etc.

I would prefer using Fig. 2 as the main figure above the abstract, instead of Fig. 1, or add some details about what the method does in the caption of Fig. 1. For instance, it’s not obvious at first glance that you are emphasizing the method’s output for different lighting settings.

Re: “In this paper, we introduce a novel task:”, this sounds a bit like you are creating an artificial problem to measure something whereas, in fact, this is a real problem that visual artists deal with, as you motivated in the introduction.

While there’s an emphasis on mesh-based video generation, from what I understand from the method specifics, the proposed approach can be used for non-mesh-based video generation as well?

It’s not clear what Fig. 8(b) refers to. The second and the sixth columns? Furthermore, I can’t really tell whether there’s a qualitative difference between those images (i.e., the blurriness).

Oh, lastly, but probably an important point, I couldn't open the videos in the supplementary. I've even tried online MP4 players in case I am missing the necessary codecs.

**Audience:**

Yes

**Audience Explanation:**

If the above given two points checks out, I believe it might be of interest to some readers, especially to those who are interested in video generation, generative games, etc.

**Broader Impact Concerns:**

I appreciate authors briefly discussed the societal impacts, and I appreciate that there is an NSFW filter built into their methods.

**Claims And Evidence:**

No

**Claims Explanation:**

I am not fully certain about how to interpret the results due to some vague points. I believe this might be addressed by explaining the metrics in detail, providing confidence intervals in each metric, and also providing some natural-looking examples, interpolations for a qualitative check. Nevertheless, in its current version, it doesn’t provide a qualitative strong evidence on its performance.

I am not sure if this should be interpreted as exploiting 3D geometry priors rather than a useful bag of tricks. I found the claim of “exploiting 3D geometry priors” is a bit strong, since it’s not clear in the paper what those priors are, and how exactly they are exploited. To me, it seems as some smart way of combining the intermediate outputs resulted in better (?) quality. I am okay with heuristics. But I don’t think this has anything to do with geometry priors. Though, please do correct me if I am wrong or missing anything.

**Requested Changes:**

**Re: exploited 3D geometry priors**

I’m not sure what 3D geometry priors this method takes into account, as claimed in the introduction. Equations 3-7 define how the method operates, but this doesn’t tell us why we should use them, and why these equations better exploit the geometric priors. For instance, even though Sections 4.2 and 4.3 are supposed to align latent vectors to generate multi-frame consistent textures, Section 4.4 tells us that this consistency might break when aggregated in the texture domain, which even contradicts the Section 4.3 title. The decrease in consistency is attributed to the view-dependent nature of depth conditions, resolution of latents, and different camera angles, which are a bit vague (e.g., isn’t the aggregation supposed to take into account all the views, and how do we even claim that it might be due to resolution of latents without any ablation on this), which makes one think whether the method indeed exploits geometric priors or just a heuristic with a motivation that works in certain conditions. For instance, in Section 4.1,

“Furthermore, we composite the rendered foreground latents with the background latents at each diffusion step (discussed in Sec. 4.2), which is essential to exploit prior in the video diffusion model (see Fig. 15).”

Why exactly is it essential? And the text continues,

“Nonetheless, such a simple aggregation method introduces blurriness in the final results.”

As we don’t exactly know the reason for the inconsistency, the proposed solution of UV blending doesn’t ground to anything—it might be helping at some level, but not very clear why this is the case. As such, I don’t think we should attribute this due to strong 3D geometric priors, as it’s not clear where those priors come into play (if they indeed). I am okay with the proposal if it is presented as “a bag of tricks that helped in such and such cases, and ablated with this metric.”

**Re: evaluation metrics challenging**

I am an outsider to this subdomain, but since we specifically consider generating textures for meshes, can’t there be a metric that compares the ground truth mesh texture with the generated textures? E.g., some sort of a game engine or a simulator where you run the model under different lighting conditions and record how you view it from different camera angles and compare it with the generated textures? Not that I’m specifically asking for this, and not that I am sure, but I thought since we’re working with mesh textures, it would have been easier to create a metric for that as we can measure how the meshes will be viewed under different conditions.

**Revising the claims, and providing natural-looking examples**

“Text-to-4D methods fail to render plausible results as they entangle geometry and appearance.”

“This issue stems from misalignment between synthesized frames and latents, where temporally consistent latents may be decoded into inconsistent RGB frames.”

“PnP-Diffusion edits frames independently and generates detailed and sophisticated appearances but suffers from poor spatio-temporal consistency due to the loss of temporal correlations in the latent space.”

I appreciate the effort in commenting on the failure cases of the methods. But I am not sure if we should attribute these failures to the mentioned causes. It would have been the least evidence to show a sequence of a video in 10-20 frames, stacking methods in rows, to show those misalignments, and show that the proposed method is better at interpolating between them. Not only there’s no such an interpolation in the main text, but the ones in the appendix have a rather weird and unnatural background (except the one with a medieval town in the background, and the example in Fig. 20, I liked that). Likewise, it’s very hard to see the temporal variations in Fig. 4, and we should have a metric for that anyways.

**Regarding metrics**

Firstly, you need multiple runs and confidence intervals.

As far as I remember, Fréchet Inception Distance requires a ground truth distribution to compare against, and based on a quick check on the Fréchet Video Distance, it seems that’s the case for FVD as well. People were using the dataset that the method was trained on to compare the distribution of generated images with the real ones. What exactly is the source distribution in the case of text-conditioned videos?

What’s the percentage in Table 1? I understand the ones in parentheses are user scores between 1 and 5.

---

> ### Author Response · Authors · 2025-09-11
>
> **Q1.** **Exploited 3D geometry priors**. Sec. 4.4 tells that consistency might break and contradict Sec. 4.2 and Sec. 4.3.
>
> **A1.** Our work focuses on dynamic texture generation, which should ensure the spatial and temporal consistency. In Sec. 4.4, the "consistency" indicates temporal consistency, and our reference UV blending module is designed to enhance the temporal consistency. While in the Sec. 4.2, the multi-view aggregation process ensure the spatial consistency, and Sec. 4.3 extends the spatial consistent textures to multiple frames. To avoid the possible misunderstanding, we have changed the title of Sec. 4.3 to "Multi-frame Texture Generation".
>
> **Q2.**  "we composite the rendered foreground latents with the background latents at each diffusion step (discussed in Sec. 4.2), which is essential to exploit prior in the video diffusion model (see Fig. 15)." why exactly is it essential?
>
> **A2.** The intuition is that the background serves as an important context component for video diffusion models to ensure the generation quality (e.g., the style or lighting/motion changes in the scene). As we discussed in the Sec. 5.3 and Fig. 8, the ablation of backgrounds would greatly deteriorate the texture quality. Therefore, our method proposes to model the foreground and background.
>
> **Q3.** Not sure what 3D geometry priors this method takes into account.
>
> **A3.** The multi-view latent aggregation operation in Sec. 4.2, 4.3 projects different views into a unified UV representation. UV is inherently based on the mesh geometry, as the faces and points in the mesh are parameterized to UV lands. In Sec 4.4, we blend latents in UV space. All modules in our method are based on the UV representation of the 3D mesh.
>
> **Q4.** Can’t there be a metric that compares the ground truth mesh texture with the generated textures?
>
> **A4.** As the generation of dynamic textures is a generative task instead of a reconstruction task, we do not have ground truth textures. The input of our method is a clay mesh (untextured mesh), which only provides the basic geometry structure, like points and faces, and no color information is included.
>
> **Q5.** Revising the claims and providing natural-looking examples.
>
> **A5.** Thanks for the suggestion. We have included examples of all methods in our supplementary video. The video is encoded with H265 due to the size limit. If your player does not support such an encoded format, we also uploaded anonymous online links, which can be directly opened in the browsers. We apologize for the inconvenience. ([Supp. Video 1](https://pub-22255ae67663459eac8663e741aa6426.r2.dev/0_Tex4D_render_demo_h265-30fps-1080p.mp4), [Supp. Video 2](https://pub-22255ae67663459eac8663e741aa6426.r2.dev/1_Tex4D_video_intro_h265-30fps-1080p.mp4)).
>
> **Q6.** You need multiple runs and confidence intervals. What’s the percentage in Table 1?
>
> **A6.** As the generation process of our method is deterministic, the confidence interval does not apply to our method. Because we conduct A/B test in our user study, which means participants are shown two methods at the same time (ours v.s. others). The percentage indicates that our method surpasses other methods in one metric.
>
> **Q7.** How is Fréchet Video Distance measured?
>
> **A7.** The original FVD is used to measure the alignment between the real video and the generated video. In our setup, we follow previous multi-view video generation tasks (e.g., VividZoo,  SV4D, and SynCamMaster) to measure FVD between videos from multiple views, which reflects the multi-view consistency in generated video clips. For the measure of multi-view consistency, the FVD metric is evaluated between pairs of all frames, which shows the semantic alignment of multi-view videos.
>
> **Q8.** What is the UV space? What is DDIM? What’s a T2I model, text-to-image?
>
> **A8.** UV mapping is the 3D modeling process of projecting a 3D model's surface to a 2D image for texture mapping. The letters "U" and "V" denote the axes of the 2D texture. UV texturing permits polygons that make up a 3D object to be painted with color (and other surface attributes) from an ordinary image. The image is called a UV texture map.
>
> DDIM stands for Denoising Diffusion Implicit Models. It’s a variant of diffusion models (like DDPM: Denoising Diffusion Probabilistic Models) but designed to make the sampling process more efficient and controllable. We have cited the original paper in our manuscript.
>
> Yes. T2I model is text-to-image model, and we have annotated the original name in the paper that first appeared.
>
> **Q9.** While there’s an emphasis on mesh-based video generation, from what I understand from the method specifics, the proposed approach can be used for non-mesh-based video generation as well?
>
> **A9.** As our method (Sec. 4.2, 4.3, 4.4) primarily builds upon the UV representation, the input mesh is necessary in our setup.

---

> ### Comment · Reviewer_Xpqh · 2025-09-25
>
> Thank you for your response. I understood the motivation behind some of the decisions. Nevertheless, you can think of my questions as rhetorical ones, and add context in the manuscript when necessary (e.g., how you compute the FVD).
>
> **Re: confidence bounds**
>
> I don't understand how the determinism of the generation process prevents you to build confidence bounds, or give an estimate of the variance. Even for estimating the FVD itself, you construct a Gaussian from a sample of generated images and compare it with a test set. You can always make a new test by generating a new set of samples? You could do the same for the A/B testing as well. E.g., GANs were deterministic (conditioned on the same noise vector) but most papers were reporting variance bounds. Let me know if I am missing something fundamental here.
>
> **Re: natural-looking examples**
> I was able to open the videos from the provided link using Firefox, thank you! The dancing figure at the beach looks quite realistic and natural-looking, though some examples are still flickering.
>
> I think my only remaining concern is about not understanding the variance in the method.

---

> ### Author Response · Authors · 2025-09-30
>
> Thanks for your response. We added the computation details of FVD in our updated PDF.
>
> **Re: variance of confidence bounds in diffusion models**
>
> We thank the reviewer for this insightful question. There is a fundamental difference in variance in GANs and diffusion-based generation. Generally in GANs, the latent $z$ is explicitly defined as a random input variable, so sampling different $z$ yields independent draws from the same conditional distribution $p(x|c)$, and confidence intervals are statistically meaningful. In diffusion, however, the initial noise is not a latent input but the starting state of a full denoising trajectory (e.g., ODE or DDIM). Changing the initial noise does not correspond to resampling from the same conditional distribution, but instead produces an entirely different denoising path. The variability that arises is largely a result of trajectory instability or hyperparameter choices, rather than statistical uncertainty around a single distribution prediction. Thus, computing confidence intervals in this way would mainly capture sampling noise rather than model reliability, and would not provide an interpretable measure of performance. Therefore, in diffusion papers, the variance (for FID) is usually not measured, and we attach some classical papers for your reference. We are ready to discuss if you have any questions.
>
>
> **Re: variance shift in our method**
>
> We provide additional clarification on the variance shift problem in our method to ensure we address any potential concerns. We discuss the variance shift problem in Sec. A.2. The DDIM denoising process assumes that the predicted noise follows $\mathcal{N}(0, \mathcal{I})$. However, multi-view aggregation disrupts this distribution. For instance, if two predicted noises $\epsilon_0 \sim \mathcal{N}(0, \mathcal{I})$ and $\epsilon_1 \sim \mathcal{N}(0, \mathcal{I})$ are aggregated into a UV map with weights $w_0$ and $w_1$ ($w_0 + w_1 = 1$), the resulting distribution becomes $\mathcal{N}(0, (w_0^2 + w_1^2)\mathcal{I})$. Since $w_0^2 + w_1^2 < 1$, the variance is reduced and no longer consistent with DDIM assumptions. Our solution is to equivalently revise the denoising formula so that all components remain consistent with the DDIM distribution. Specifically, we first compute and aggregate the predicted original samples, which are expected to have 0 variance under DDIM, ensuring that the aggregated result $\hat{\mathcal{T}}_0$ also has zero variance. We then denoise the UV map starting from an initial Gaussian conditioned on $\hat{\mathcal{T}}_0$, thereby guaranteeing a correct and consistent update.
>
> **Reference:**
>
> Patrick Esser, et al. "Scaling Rectified Flow Transformers for High-Resolution Image Synthesis." arXiv preprint arXiv:2407.17470 (2024).
>
> Robin Rombach, et al. "High-Resolution Image Synthesis with Latent Diffusion Models." arXiv preprint arXiv:2112.10752 (2021).
>
> Wenliang Zhao, et al. "UniPC: A Unified Predictor-Corrector Framework for Fast Sampling of Diffusion Models." arXiv preprint arXiv:2302.04867 (2023).
>
> Jiaming Song, et al. "Denoising Diffusion Implicit Models." arXiv preprint arXiv:2010.02502 (2020).
>
> Cheng Lu, et al. "DPM-Solver++: Fast Solver for Guided Sampling of Diffusion Probabilistic Models." arXiv preprint arXiv:2211.01095 (2022).

---

### Review · Reviewer_3i3m · 2025-08-27

**Summary Of Contributions:**

## Summary

This paper proposes a zero-shot approach for a novel task: texturing 3D character mesh sequences given text prompts. The method leverages a depth-conditioned video diffusion model and introduces a DDIM sampling procedure to directly update UV maps of the character mesh across multiple frames and camera views. The DDIM sampling strategy aggregates noise updates from different projected views to produce temporally coherent UV maps.

## Strengths

1. **Novel Task**. The paper addresses a new and interesting problem: generating textures for dynamic 3D meshes from text prompts.

2. **Zero-shot capability**. The proposed method requires no 3D data for training. Instead, it exploits the strong generative priors of video diffusion models and generates UV maps directly. This zero-shot property is particularly valuable for 3D domains, where collecting and curating datasets is costly. In addition, the approach is feedforward, which makes it significantly faster than optimization-based methods such as SDS.

3. **Novel technical design**. To enable noise updates on UV maps, the authors propose an intuitive strategy to aggregate updates from multiple overlapping camera projections. Other technical contributions, such as separate foreground/background masking for feature integration, are also interesting and well-motivated.

4. **Background generation**. Beyond mesh texture generation, the method can also produce dynamic background videos, which enhances its applicability.

5. **SoTA performance**. Qualitative comparisons suggest that the proposed approach achieves superior visual results compared to existing baselines.

## Weaknesses

1. **Dynamic vs. static texture**. Traditionally, texture is a static property of a mesh, ensuring consistency across environments. In contrast, this work assigns a separate texture to each frame, effectively producing “dynamic” or “4D” textures. The motivation for this design choice is not clearly articulated, and the benefits over static textures remain unclear. Moreover, this design can easily lead to temporal inconsistencies.

2. **Consistency issues**. Generated textures exhibit noticeable flickering artifacts. Some textures change abruptly at specific frames, while others drift continuously over time (e.g., the snowman example). This weakens the overall realism.

3. **Limited fidelity and detail**. The textures often lack fine-grained details, resulting in oversimplified or flat appearances. For instance, the snowman and the dancing robot are rendered with only basic color schemes and minimal structure.

4. **Experimental evaluation**. The experimental section could be stronger. While the authors compare against several baselines, the evaluation scale and methodology are not clearly described. In particular, the number of samples, the procedure for generating baseline results, and the representativeness of these samples are not specified. The reliance on user study raises concerns about statistical robustness. Furthermore, the use of FVD as a metric requires more explanation, especially regarding its relevance for evaluating texture quality.

**Additional Comments:**

I'm not an expert in 3D mesh/shape modeling. Therefore, my evaluation is more high-level, not on the technical side.

**Audience:**

Yes

**Audience Explanation:**

This work focuses on generating mesh textures and will be of interest to researchers in 3D generative modeling.

**Broader Impact Concerns:**

I don't see broader impact concerns.

**Claims And Evidence:**

Yes

**Claims Explanation:**

The claims made in the submission are mostly met in their latter method designs. First, their work is indeed a zero-shot pipeline for 4D texture generation. To this end, they utilize video diffusion models, and modify the DDIM sampling process to realize their goal. Since their model is feedforward approach, it is also computationally efficient.

**Requested Changes:**

1. **Clarify motivation for dynamic textures.** - Explain why generating per-frame (dynamic/4D) textures is necessary and important. An ablation study on this design choice is favorable.

2. **Strengthen experimental evaluation.**
   - Clearly specify the evaluation setup:
     - Number of test samples.
     - How baseline results were generated.
     - Whether samples are representative and fairly chosen.
   - Expand beyond user studies:
     - Include quantitative evaluations on texture quality (if possible).
     - Provide stronger justification for the use of FVD and explain why it is suitable for texture evaluation.
   - If applicable, consider larger-scale or more diverse experiments to validate robustness.

3. Since the authors claimed advantage on computation efficiency, runtime comparions are necessary.

---

> ### Author Response · Authors · 2025-09-11
>
> **Q1.** **Clarify motivation for dynamic textures.** Explain why generating per-frame (dynamic/4D) textures is necessary and important.
>
> **A1.** We appreciate the reviewer’s question. As discussed in the introduction, dynamic textures are essential for modeling effects that static textures cannot easily capture, such as lighting changes, view-dependent reflections, or character appearance transformations. Unlike static textures, which require complex lighting control to simulate these effects, dynamic textures encode them directly at the per-frame level (Fig. 2, 12). Moreover, in modern graphics pipelines, dynamic textures are a fundamental building block for producing vivid, temporally coherent videos, underscoring both the necessity and significance of studying dynamic (4D) texture generation. Additionally, we emphasize the advantage of dynamic textures (ours) with static textures in Fig 7. *Text2Tex* shows empty texels in invisible regions and fails to model dynamics.
>
>
>
> **Q2.** **Strengthen experimental evaluation.** Clearly specify the evaluation setup. Provide stronger justification for the use of FVD and explain why it is suitable for texture evaluation.
>
> **A2.** Thank you for this suggestion. We provide the full evaluation setup in Sec F.6, which includes: (1) the test samples used for FVD and consistency evaluation,  (2) detailed protocol of the user study. These details will also be clarified in the revision to ensure transparency and reproducibility, and (3) Details of Baseline Methods.
>
> - Specifically, our test set for the FVD metric contains 20 distinct mesh sequences, including 7 non-human characters (from Sketchfab) and 13 human-like characters (from Mixamo and human diffusion models). This balance ensures diversity across both geometry and character type. To increase variation, we assign multiple stylization prompts per sequence, resulting in a total of 32 test cases. This setup allows evaluation of both generalization across diverse meshes and controllability under varied textual instructions.
>
> - Our baseline comparisons include (i) zero-shot video generation methods, (ii) zero-shot texture generation methods, and (iii) 4D generation methods. We clarify the experimental settings for each baseline in Sec. A.4.1.
>
>
> - FVD is a standard metric in video generation tasks (e.g., VividZoo, SV4D, and SynCamMaster). Since our comparisons include zero-shot video generation models, and our task focuses on **spatial and temporal consistency**, FVD is a natural choice to evaluate **temporal coherence**. While not texture-specific, it is widely recognized for capturing frame-to-frame smoothness, making it a suitable metric for dynamic texture evaluation.
>
> **Reference:** Xie, Yiming, et al. "SV4D: Dynamic 3d content generation with multi-frame and multi-view consistency." arXiv preprint arXiv:2407.17470 (2024).
>
>
> **Q3.** Since the authors claimed advantage on computation efficiency, runtime comparions are necessary.
>
> **A3.** We apologize for the earlier confusion. Our claim of computational efficiency refers to the fact that our method is **zero-shot**: it requires no model training or fine-tuning, making it significantly lighter than training-based baselines. To avoid misinterpretation, we have revised the statement in the last sentences of introduction to emphasize that our approach is a **plug-and-play solution** rather than presenting direct runtime speedups.

---

### Review · Reviewer_N3xW · 2025-08-28

**Summary Of Contributions:**

The paper introduces the novel problem of zero-shot 4D character texturing, formulating the task of generating temporally and spatially consistent dynamic textures directly from text prompts without requiring paired training data. To address this, the authors propose a UV-space aggregation framework that ensures global surface coherence by projecting and combining multi-view latent features within a consistent parameterization domain, thereby avoiding the discontinuities that typically arise from view-dependent synthesis. Furthermore, the work identifies and analyzes the variance shift issue caused by multi-view latent aggregation, and presents a variance-compensated DDIM reformulation in the UV domain to restore correct noise dynamics during sampling. The method is further enhanced through a foreground–background decomposition strategy that introduces learnable background latents, as well as a reference-UV blending mechanism that improves temporal stability and effectively handles occlusions by progressively filling missing texels across frames. Finally, the approach is validated through extensive experiments and user studies, showing clear improvements in both spatial consistency and temporal coherence over state-of-the-art text-to-texture and text-to-video baselines.

**Audience:**

Yes

**Audience Explanation:**

The topic of zero-shot 4D texturing with video diffusion models is timely and relevant to the TMLR audience, especially those interested in generative modeling, computer vision, and graphics. The paper introduces a novel formulation that bridges video diffusion models and dynamic 3D/4D content creation, which has clear implications for areas such as animation, gaming, and virtual reality. The methodological innovations, including UV-space aggregation and variance-compensated sampling, are of broad interest to researchers exploring cross-modal generative techniques and consistency in spatiotemporal generation.

**Broader Impact Concerns:**

The work could raise some broader impact concerns. In particular, the ability to synthesize realistic 4D character textures may be misused for generating deceptive or unauthorized content. A short discussion acknowledging these potential risks would make the paper more complete.

**Claims And Evidence:**

Yes

**Claims Explanation:**

The claims made in the submission are generally well supported by both qualitative and quantitative evidence. The authors present extensive comparisons against strong baselines, including Text2Tex, Text2Video-Zero, PnP-Diffusion, TokenFlow, and others, and demonstrate consistent improvements across multiple metrics such as FVD and CLIP-based Consistency Scores. In addition, ablation studies carefully isolate the impact of each proposed component (UV aggregation, variance-compensated DDIM, background priors, and reference-UV blending), and user studies further validate the perceptual advantages of the method. While some aspects, such as the theoretical analysis of variance shift, could benefit from more detailed quantification, the experimental evidence provided is convincing, clear, and sufficient to support the main contributions and conclusions.

**Requested Changes:**

1. While the UV-domain DDIM reformulation is intuitively reasonable, the paper does not provide sufficient quantitative or theoretical evidence that it fully compensates for the variance reduction introduced by multi-view aggregation. A more formal derivation or empirical measurements across timesteps are necessary to convincingly support this claim.

2. In several cases, such as the comparison with Generative Rendering, the results are drawn from demos rather than controlled experiments, which weakens the strength of the evidence. The paper should clearly specify the experimental settings, input conditions, and limitations for each baseline to ensure transparency and reproducibility.

3. The evaluation could be strengthened by incorporating geometry-aware metrics. While FVD and CLIP-based Consistency Scores are useful, they do not fully capture UV-space or reprojection consistency, which are central to the problem addressed.

4. The influence of key hyperparameters, such as the blending weight λ, the cosine power α, and the keyframe interval, is not systematically reported.

5. The paper should provide more discussion on scalability and robustness over long sequences. The authors acknowledge that quality degradation occurs when generating very long sequences, but the analysis remains limited.

---

> ### Author Response · Authors · 2025-09-11
>
> **Q1.** UV-domain DDIM reformulation does not provide quantitative or theoretical evidence. A more formal derivation or empirical measurements across timesteps are necessary to convincingly support this claim.
>
> **A1.** We appreciate the reviewer’s feedback. To strengthen our argument, we have included a formal derivation in Eq. (8), Sec. A.2, which explicitly characterizes the latent variable $z$ at timestep $t$. This provides theoretical support for our claim of "variance shift."
>
>
>
> **Q2.** `Generative Rendering` results are from demos. The paper should clearly specify the experimental settings, input conditions, and limitations for each baseline to ensure transparency and reproducibility.
>
> **A2.** Thank you for raising this important point. Since dynamic textures are primarily used for **dynamic video creation**, our comparisons include **(i) zero-shot video generation methods, (ii) zero-shot texture generation methods, and (iii) 4D generation methods**. We clarify the experimental settings and limitations as follows (added to Sec. A.4.1):
>
> **Zero-shot video generation methods:**
>
> 1. **PnP-Diffusion.** Originally designed for image-to-image stylization, we adapt it for video by following TokenFlow: each frame of the clay-rendered mesh sequence is stylized independently. This simple extension enables intuitive zero-shot stylization, but the lack of temporal constraints leads to strong flickering artifacts across frames (as shown in the supplementary video).
>
> 2. **TokenFlow, Text2Video-Zero, and LatentMan.** These models perform zero-shot video editing by directly modifying the latent codes of diffusion models. In our setup, we first render an untextured animation video, then apply these models with textual prompts to generate stylized videos. However, because diffusion latents are highly view-sensitive, these methods fail to preserve multi-view consistency, resulting in textures that vary across different camera viewpoints.
>
> 3. **Gen-1.** We also include *Gen-1*, a commercial model that supports untextured video stylization. Following the protocol in TokenFlow, we apply it to our clay-rendered animation inputs. While it improves stylization quality compared to open-source models, it still struggles with temporal flickering and lacks explicit handling of multi-view coherence.
>
>
>
> **Zero-shot texture generation methods:** Dynamic textures provide two advantages compared to static textures: (1) they naturally capture motion patterns, and (2) they mitigate untextured regions caused by occlusions, since an occluded area in one frame may appear in another. We evaluate *Text2Tex* (Fig. 7) as a representative method. However, because it only inpaints textures in the **canonical space**, it frequently produces **empty or incomplete textures**, especially on regions unseen in the canonical view.
>
>
>
> **4D Generations:** Although not zero-shot, we include 4D generation approaches for completeness. These methods take an untextured mesh and a textual prompt as input, similar to our setup. However, they struggle with multi-view consistency due to their reliance on generative rendering pipelines rather than explicit UV-based modeling.
>
>
>
> **Q3.**  The evaluation could be strengthened by incorporating **geometry-aware metrics**. While FVD and CLIP-based Consistency Scores are useful, they do not fully capture UV-space or reprojection consistency, which are central to the problem addressed.
>
> **A3.** We agree with the reviewer that geometry-aware metrics could reflect the spatial conherence of different methods, and we have followed *SyncMVD* to incorporate a **3D-consistency score**, which is explicitly geometry-aware. Specifically, the 3D-consistency score of generated textures is computed by averaging similarities between all rendered/generated views (in our case, 7 views in total), to reflect the extent of geometric consistency.
>
>
>
> **Q4.** The influence of key hyperparameters, such as the blending weight λ, the cosine power α, and the keyframe interval, is not systematically reported.
>
> **A4.** To systematically report the hyperparameters, we provide the results in the Fig. 22. Our method is generally robust under different key hyperparameters.
>
> **Q5.** The paper should provide more discussion on scalability and robustness over long sequences. The analysis remains limited.
>
> **A5.** Thank you for pointing this out. We have expanded our discussion in the revision to better analyze the scalability and robustness of Tex4D for long texture sequence generation (Sec. F.4). We further report the CLIP metric at the same intermediate frame (100th) to quantify the deteriorated effects with long video sequences. This is consistent with prior observations on video diffusion models, which tend to distort high-frequency details over extended generations (VideoCrafter2).

---

### Review · Reviewer_9ymm · 2025-08-31

**Summary Of Contributions:**

This paper introduces a method for zero-shot 4D character texturing by leveraging a text-to-video diffusion model. The approach takes a sequence of untextured meshes as input and outputs temporally and multi-view consistent textured meshes guided by text instructions. The pipeline first renders images and depth maps of the mesh sequence from K different viewpoints. For each viewpoint, these renderings are passed through the video diffusion model, CTRL-Adaptor, to generate temporally consistent textures. To enforce multi-view consistency, the method aggregates textures from different views into a single UV map and applies diffusion-based denoising in UV space. Additionally, the authors propose learning the background separately and incorporating a reference UV map to further enhance performance. Experimental results show that the method outperforms existing approaches, achieving superior results in 4D mesh texturing.

**Audience:**

Yes

**Audience Explanation:**

Yes, I believe the work of this paper is valuable to some readers of TMLR. This paper works on texturing 4D meshes using the video diffusion in a zero-shot manner. The combination of the diffusion models and 4D meshes is a popular research direction and a significant challenge in computer vision. Therefore, certain readers will be interested in the work of this paper.

**Broader Impact Concerns:**

No perceived ethical implications of this work.

**Claims And Evidence:**

Yes

**Claims Explanation:**

The main contribution of this paper is the proposal of a method for zero-shot 4D mesh texturing using video diffusion models. After reading the submission, I find this claim largely substantiated, though some areas require further clarification. My assessment is as follows:

**Strengths**
1. **Clear problem identification and motivation**: The introduction clearly articulates the limitations of existing methods. Using video diffusion models directly leads to multi-view inconsistency, while relying on static text-to-3D diffusion models results in temporal inconsistency. The paper correctly identifies that 4D mesh texturing requires solving both inconsistencies simultaneously, establishing a strong motivation.
2. **Intuitive and logical method design**: The proposed approach is well aligned with the identified challenges. Rendering mesh sequences from multiple viewpoints and first using video diffusion to achieve temporal consistency before aggregating across views for multi-view consistency is both intuitive and methodologically sound.
3. **Comprehensive evaluation**: The method is benchmarked against a wide range of baselines, including both video diffusion and static text-to-3D approaches. The evaluation employs two objective metrics (FVD and CLIP-based Consistency Score) alongside a large-scale user study involving 67 participants, which is notable in comparison to related works. Ablation studies further demonstrate the contribution of individual components, strengthening the overall empirical evidence.

**Weaknesses**
1. **Unclear explanation of the proposed solution to variance shift**: In Section 4.3, the paper notes that a “simple aggregation and projection strategy leads to a blurry appearance” and attributes this to a variance shift caused by texture aggregation. However, the explanation of how the proposed solution addresses this issue is insufficient. From my understanding, v-parameterization is simply an alternative to the epsilon formulation, and the variance shift in epsilon would still be present in v-parameterization. The paper does not convincingly explain why this change resolves the variance shift problem.
2. **Limited dataset details**: The description of the dataset is sparse. More information is needed regarding the number of mesh sequences, diversity of scenes, and the number of texture instructions per scene. These details are important for assessing the scale, diversity, and generalizability of the method.
3. **Ambiguities in the user study description**: Section 5.2 states: “Our study involved 67 participants who provided a total of 1104 valid responses based on six different scenes drawn from six previous works, with each scene producing videos from two different views.” The meaning of “six different scenes drawn from six previous works” and “each scene producing videos from two different views” is unclear. More precise explanation of the study setup would strengthen the validity and reproducibility of the evaluation.

Despite some ambiguities and unclear explanations, the submission provides substantial evidence supporting its main claim. The method is well motivated, logically designed, and empirically validated with both quantitative metrics and a large-scale user study.

**Requested Changes:**

1. Please explain how the formulation in Section 4.3 addresses the “variance shift” caused by texture aggregation.
2. Please provide more information regarding the scale and the diversity of the dataset.
3. Please provide more information regarding the setup of the user study.
4. Please change a different letter for the output of the UNet in equations. The $\epsilon_{\theta}$ cause significant confusion with the $\epsilon_{\theta}(z_t)$ when reading the equation.

---

> ### Author Response · Authors · 2025-09-11
>
> **Q1.** **Unclear explanation of the proposed solution to variance shift**. The explanation of how the proposed solution addresses this issue is insufficient.
>
> **A1.** We appreciate the reviewer’s suggestion. To address this concern, we have added a formal proof about the variance shift problem in Eq. (8), Sec. A.2, which explicitly shows how the latent variable $z$ evolves at timestep $t$. We hope this proof provides a rigorous foundation for our claim of "variance shift", and then we discuss how our proposed solution effectively mitigates this issue.
>
> **Q2.** **Limited dataset details**. The description of the dataset is sparse. More information is needed regarding the number of mesh sequences, the diversity of scenes, and the number of texture instructions per scene.
>
> **A2.** Thank you for highlighting this point. For FVD and consistency score evaluation, our test set consists of **20 distinct mesh sequences**. To ensure diversity, we intentionally balanced human and non-human characters: **7 out of 20 sequences are non-human characters**, primarily sourced from Sketchfab, while the remaining 13 are human-like characters drawn from Mixamo and human diffusion models. This design prevents the dataset from being biased toward human characters and encourages evaluation on more challenging and varied shapes.
>
> We have added the dataset details in Sec. A.4.
>
> To further enhance diversity, each mesh sequence is paired with multiple **textual prompts for stylization**, randomly allocated to introduce variation in texture instructions. This process yields a total of **32 test cases**. By combining human-like and non-human meshes with varied prompts, our dataset enables a comprehensive evaluation of both generalization (across different geometries) and controllability (across different style instructions).
>
>
>
> **Q3.** Please provide more information regarding the setup of the user study.
>
> **A3.** We conducted the user study by comparing 6 different methods (Tab. 1, including ours). Each method was evaluated on 8 distinct scenes (mesh sequences with textual prompts), resulting in 6x8 = 48 multi-view video outputs for assessing spatial and temporal coherence. The user study followed an A/B testing protocol (Fig. 18): participants were asked to (1) choose the better result between two methods and (2) rate each metric on a scale of 1–5. To ensure fairness and clarity, for each case (ours v.s. other method) we presented two views, randomly selected from 7 predefined camera viewpoints, and composited them into a multi-view video. The user study involves 67 participants who provided a total of 1104 valid responses.
>
> We added the details of the user study and baseline methods in Sec. A.4.
>
> **Q4.** Please change a different letter for the output of the UNet in equations. The $\epsilon_\theta$ causes significant confusion with the $\epsilon_\theta(z_t)$ when reading the equation.
>
> **A4.** Thanks for the suggestion, we have revised $\epsilon_\theta$ to $\epsilon_{\text{UNet}}$ in our manuscript (Eq. 1, 2, Sec. 3).

---

### Decision · Action_Editor_8wNT · 2025-10-08

**Recommendation:** Reject

**Additional Comments:**

NA

**Audience:**

Yes

**Audience Explanation:**

Chatecter texturing as an promising usage of diffusion models, is important to the area of new media. The idea proposed in this submission is interesting and inspiring, and the initial experiments show some suprior results, whereas some parts in terms of some small modules, experimental design and metrics to demonstrate and explain the performance can be further improved.

**Claims And Evidence:**

No

**Claims Explanation:**

This submission studied 4D character texturing using video diffusion models, which leverages a sequence of untextured meshes and text prompt as input and incorporates the geometry knowledge inherent in video diffusion models to build multi-view and temporally consistent textured meshes.  To enforce the consistency, the authors carefully manipulate the sampling process with the prior knowledge guidenace and enhancement. Experimental results show that the method outperforms existing approaches, achieving superior results in 4D mesh texturing.

The submission received the reviews of four reviewers, who finally recommended "Leaning Reject" * 3 and "Leaning Accept" * 1 after the one-round reviewer-author interaction. Generally, the reviewers' concerns remain surrounding the following points:

1) The method brings some concerns about the performance according to the generated textures, many of which lack details. This is mainly because the authors follow a dynamic texture setting in technical design, which to some extent induces some drawbacks in performance.

2) About the experiments that are based on 20 samples, this is, to some extent, unconvincing in realist cases, since it is not easy to guarantee their representiveness and statistical conficence.

3) About the metrics used to evaluate the sample quality and variance, it is important to characterize the confidence bounds and the details about the user study. Although the author response provided some information about the user study, the former one remains as a critical concern to the reviewer, which has not been well addressed.

Overall, the idea and some technical design is interesting and good, however, the performance and their experimental basis as well as the adopted metrics are still not very convincing to reach the consensus to accept the submission. AE tends to recommend "Leaning Reject", and hope the  concerns and suggestions help improve the submission in the future.